# SUGARCREPE++ Dataset: Vision-Language Model Sensitivity to Semantic and Lexical Alterations

**Sri Harsha Dumpala**[*]    **Aman Jaiswal**[*]    **Chandramouli Sastry**
**Evangelos Milios**    **Sageev Oore**    **Hassan Sajjad**
Dalhousie University, Canada.

## Abstract

Despite their remarkable successes, state-of-the-art large language models (LLMs), including vision-and-language models (VLMs) and unimodal language models (ULMs), fail to understand precise semantics. For example, semantically equivalent sentences expressed using different lexical compositions elicit diverging representations. The degree of this divergence and its impact on encoded semantics is not very well understood. In this paper, we introduce the SUGARCREPE++ dataset to analyze the sensitivity of VLMs and ULMs to lexical and semantic alterations. Each sample in SUGARCREPE++ dataset consists of an image and a corresponding triplet of captions: a pair of semantically equivalent but lexically different positive captions and one hard negative caption. This poses a 3-way semantic (in)equivalence problem to the language models. We comprehensively evaluate VLMs and ULMs that differ in architecture, pre-training objectives and datasets to benchmark the performance of SUGARCREPE++ dataset. Experimental results highlight the difficulties of VLMs in distinguishing between lexical and semantic variations, particularly to object attributes and spatial relations. Although VLMs with larger pre-training datasets, model sizes, and multiple pre-training objectives achieve better performance on SUGARCREPE++, there is a significant opportunity for improvement. We demonstrate that models excelling on compositionality datasets may not perform equally well on SUGARCREPE++. This indicates that compositionality alone might not be sufficient to fully understand semantic and lexical alterations. Given the importance of the property that the SUGARCREPE++ dataset targets, it serves as a new challenge to the vision-and-language community. Data and code is available at `https://github.com/Sri-Harsha/scpp`.

## 1 Introduction

Large language models (LLMs), including vision-and-language models and unimodal language models, have shown tremendous results in solving a majority of vision and natural language processing (NLP) tasks. Surprisingly, despite such success, LLMs can exhibit different behaviors for semantically equivalent sentences composed with different syntactic or lexical structures. Previous works have reported such lack of compositional reasoning in both vision-and-language models (e.g., [83, 97, 100, 69, 87]) and unimodal language models (e.g., [40, 11, 104, 57]). For instance, the performance of the state-of-the-art (SOTA) LLMs including GPT-4, Gemini and Llama are sensitive to the prompt formatting [77]. The model editing techniques [11, 57] suffer from misfired edits due to the dominance of lexical overlap [90, 71]. Similarly, Zou et al. [104] demonstrated that safety aligned models can be "jailbroken" by simply appending an adversarial suffix causing them to generate objectionable content bypassing all safeguards.

---

[*] The authors contribute equally to this work.

38th Conference on Neural Information Processing Systems (NeurIPS 2024) Track on Datasets and Benchmarks.

These observations suggest that language models' perception of semantic similarity crucially depends on the lexical representation of the sentence and calls for a stricter evaluation of semantic text similarity that factors in lexical and syntactic structures. Semantic text similarity is one of the oldest metrics to evaluate language understanding [92, 65, 84, 28, 27, 33, 79] and despite recent evidence of lexical sensitivity, large benchmarks (e.g., [80, 61]) evaluate semantic similarity without explicitly considering the lexical influence. In this work, we aim to address this gap by proposing a dataset to perform joint evaluation of semantic understanding — through the semantic equivalence detection task (elaborated below) — and lexical sensitivity in language models.

Recognizing semantic similarity is often viewed as being fundamental to language understanding. In fact, strong performance on semantic text similarity is often predictive of a language model's performance in various downstream applications [12] including question-answering, retrieval and summarization. Based on overlap in their meaning, a pair of sentences can be roughly labeled as semantically equivalent, semantically similar or semantically dissimilar. More specifically, semantically equivalent sentences convey the *same meaning* [2], perhaps differing in terms of syntactic [3] and lexical [4] structures. On the other hand, sentences that are not semantically equivalent but describe the *same topic* are said to be semantically similar [3]. Important examples include MRPC [20], QQP [34] and STS [2–6]: while MRPC and QQP contain binary labels indicating semantic equivalence, STS uses a score between 0 to 5 to indicate the degree of semantic equivalence.

The timely release of these datasets have fueled the research and development of improved language models. While these datasets remain relevant even today and are included as part of the challenging GLUE benchmark [85], we aim to improve upon the following aspects to evaluate language understanding through ***semantic equivalence task under controlled lexical constraints***:

(a) Varying Definitions of Semantic Equivalence: While MRPC [20] aims to evaluate if language models can detect semantic equivalence, it ultimately uses a loose definition of equivalence in that the sentence pairs that convey different information about the same topic are also considered to be semantically equivalent. Different from semantically equivalent sentences, a pair of questions are defined to be semantically equivalent if they have the same answer [72] and hence, datasets on semantically equivalent question pairs (e.g., QQP) require additional knowledge beyond language understanding. In contrast, we focus our evaluation on fundamental language understanding ability and evaluate a language model in terms of its ability to recognize semantic equivalence between a pair of sentences. In this work, two sentences are said to be semantically equivalent if the sentences convey the same meaning and can be inferred from each other (i.e., bidirectional entailment).

(b) Lack of Lexical Constraints: While achieving perfect scores on the existing semantic similarity datasets is indeed challenging, trivial baselines using lexical overlap also provide reasonable estimates of semantic similarity (for example, see Figure 2 and Table 6 in Abdalla et al. [1]). Therefore, the extent to which language models rely upon lexical structure when identifying semantic equivalence and semantic similarity is not clearly known. We are thus motivated to explore a more challenging setting that requires a language model to encode semantics beyond superficial lexical structure.

Closer to our goal, Hsieh et al. [29] introduced the challenging SUGARCREPE dataset to evaluate the ability of vision-language models (VLMs) to identify *one* correct caption in a pair of lexically similar sentences. As an example, given an image, the model may be asked to select the correct caption between "*A tractor and two boats far from the water*" (incorrect) and "*A tractor and two boats beside the water*"(correct). Hsieh et al. [29] reported that several VLMs face challenges in selecting the correct caption and attributed the low performance to the text encoder's inability to identify semantic differences in a pair of lexically similar sentences. While this dataset provides a good starting point, it is insufficient for comprehensively evaluating the lexical sensitivity and semantic understanding of a model; the model's understanding of semantic equivalence in the presence of lexical differences remains unclear from such an evaluation. For instance, "*Couple of boats and a tractor located next to the water*" is semantically identical to the correct caption despite lexical dissimilarities. A precise evaluation of lexical influence upon semantic understanding *should* include

---

[2]*Semantic* [59] relates to meaning in language.

[3]*Syntax* [60] refers to the way in which linguistic elements (such as words) are put together to form constituents (such as phrases or clauses).

[4]*Lexical* [58] relates to the words or the vocabulary of a language as distinguished from its grammar and construction.

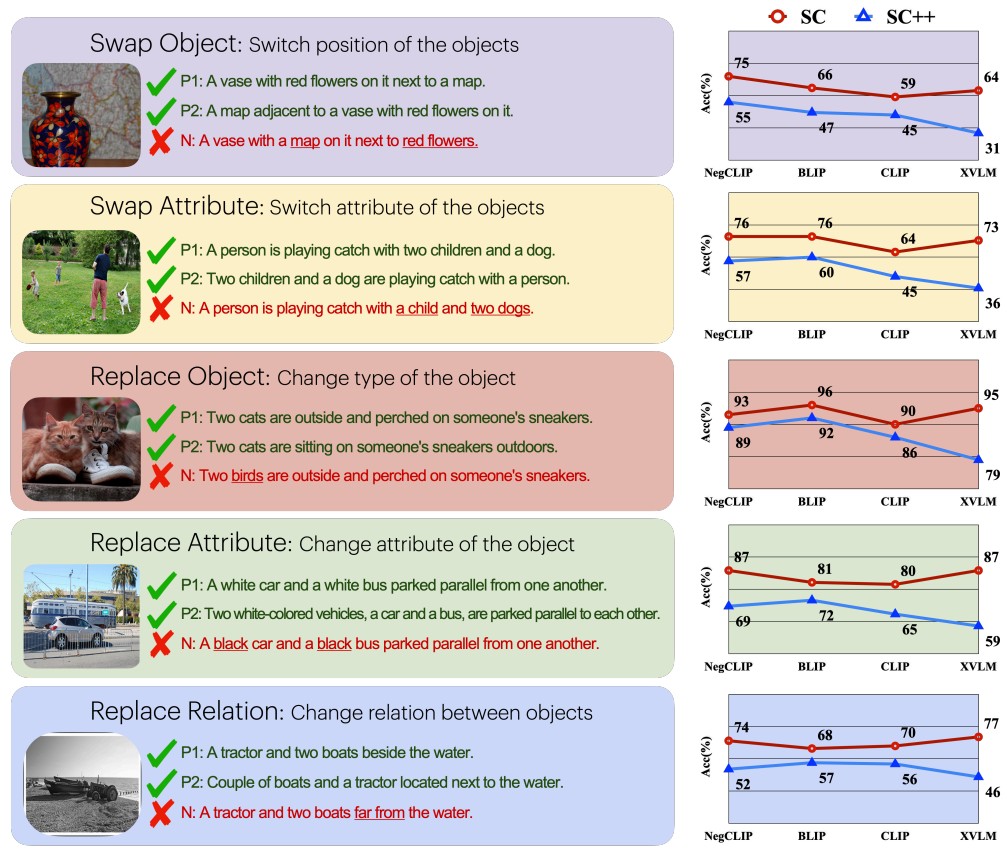

Figure 1: Examples from SUGARCREPE++ (SC++) dataset. $P_1$ and $P_2$ are semantically equivalent but lexically different while $N$ is semantically different than both $P_1$ and $P_2$ despite its lexical similarity with $P_1$. The adjacent line charts highlight the performance gaps in VLMs discovered upon re-evaluation using SC++, and shows that strong lexical and semantic understanding may not be required to achieve better performance on SUGARCREPE (SC). Refer to Appendix C.4 for details on negative captions (e.g., swap-object, replace-relation).

pairs of semantically-equivalent, semantically-opposite, lexically-similar, and lexically-dissimilar sentences. In this work, we target these four cases and define the following research questions:

1. How well do LLMs understand the *semantic **equivalence*** between a pair of sentences given their *syntactic and lexical **differences***?
2. How well do LLMs understand the *semantic **differences*** between a pair of sentences given their *syntactic and lexical **similarities***?

To that end, we extend SUGARCREPE to introduce SUGARCREPE++ dataset which additionally contains semantically equivalent sentences that are lexically dissimilar. The answer to these two questions enable us to evaluate the semantic understanding while disentangling the effect of lexical matches between sentences.

Our contributions to this work are as follows:

- **SUGARCREPE++ dataset.** We introduce SUGARCREPE++, a diverse, multi-modal and human-validated dataset, especially designed to evaluate the sensitivity of encoded semantics in language models to lexical composition. The introduction of a three-way semantic (in)equivalence task enables evaluation with increased resolution previously not possible with two captions[29]. Figure 1, illustrates instances from five categories incorporated in SUGARCREPE++ and highlights the apparent gaps in performance of VLMs when re-evaluated using SUGARCREPE++.
- **Unified evaluation.** We designed SUGARCREPE++ dataset such that the overlap of semantic information between the two positive captions is always higher than between the positive and negative captions, even without considering the image. This allows us to evaluate on the Text-to-Text task twice for the same triplet, using each positive caption as the reference once.

These caption triplets combined with an image enables a previously unexplored dual-mode evaluation of VLMs in both Image-to-Text and Text-to-Text settings.

In this work, we evaluate a comprehensive list of VLMs and standalone/unimodal language models (ULMs) using SUGARCREPE++. A few of the notable findings are summarized below:

- VLMs struggle to identify the differences between semantic and lexical alterations, particularly if the lexical alterations are based on swapping attributes or objects, or replacing relations.
- There exists a large gap between the VLMs and human-level performance signifying huge scope for improvement in VLMs.
- Text encoders of VLMs form the major bottleneck in achieving better performance on SUGARCREPE++.
- Even state-of-the-art ULMs fail to consistently dissociate semantics from lexical forms i.e., they tend to select captions with higher lexical overlap than the ones with higher semantic overlap.

## 2 SUGARCREPE++

In this section, we describe the data generation and validation pipeline used to create SUGARCREPE++ dataset. The SUGARCREPE dataset [29] — derived from MS-COCO [49], a dataset of image-caption pairs — consists of a correct caption and an incorrect caption for each image while ensuring that the two captions are lexically similar. To create SUGARCREPE++ dataset, we generate another *correct* caption for each image such that it uses alternative lexical representation while being semantically identical to the original correct caption in SUGARCREPE. If we refer to the correct captions as positives, the incorrect caption can be termed as a *hard negative* due to its high lexical overlap with one of the positive captions. In contrast to SUGARCREPE, SUGARCREPE++ enables evaluation across both multimodal and unimodal settings while also providing a comprehensive evaluation of semantic understanding with lexical constraints. Additional related work is discussed in Appendix B.

### 2.1 Dataset Generation and Validation

Prior works [8, 24, 41, 64] have extensively used image-caption pairs from MS-COCO; specifically Crepe [56] and its improved derivative SUGARCREPE [29] that leverages the recent advancements in conditional text generation using large language models (LLMs) to generate hard negative captions, thereby overcoming the issues with procedurally generated captions. The SUGARCREPE dataset consists of (only) one positive and one hard negative caption for each image. Relative to the negative caption, a single positive caption can either have low or high lexical overlap. The original SUGARCREPE only captures the high overlap case. To evaluate the sensitivity of encoded semantics to lexical alteration, we require an additional positive caption with a different lexical composition. We build SUGARCREPE++ using instruction fine-tuned Mistral 7B [36] model to further introduce an additional positive caption as illustrated in Algorithms 1 and 2. The generation process of the additional caption can be divided into two stages: (1) generation using meta-prompts and (2) automated and human validation of generated captions.

---

**Algorithm 1** Generation Pipeline

**Require:** Original caption $P_1$
1: $P_2 \leftarrow LLM(M\{P_1\})$
2: Accept $\leftarrow False$
3: **while not** Accept **do**
4:     **if** Duplicated$(P_1, P_2)$ **then**
5:         $P_2 \leftarrow LLM(M\{P_1; P_2\})$
6:     **else if** Superfluous$(P_2, P_1)$ **then**
7:         $P_2 \leftarrow LLM(M\{P_1\})$
8:     **else**
9:         $Accept \leftarrow True$
10: **return** $P_2$

---

**Algorithm 2** Automatic Validation Pipeline

**Require:** Original caption $P_1$,
    Generated caption $P_2$
1: $Valid \leftarrow LLM(V\{P_2, P_1\})$
2: **while not** Valid **do**
3:     $P_2 \leftarrow GenerationPipeline(M\{P_1\})$
4:     $valid \leftarrow LLM(V\{P_2, P_1\})$
    **Return** $P_2$

---

> **Role-playing Prompt:** You are an instruction-following DataGenAI. Your expertise lies in accurately interpreting and generating datasets based on given instructions. Your responses are expected to be precise and limited to the required output.

Figure 2: Role playing prompt for "Data Generator AI".

**Generation using meta-prompts:** Algorithm 1 presents our generation pipeline. We followed an iterative prompting methodology to refine a meta-prompt (M) that generates optimal second positive captions ($P_2$) given the original positive caption ($P_1$). The meta-prompt (M) is a composition of constituent prompts ($M_i$) formed by concatenating (;) different sub-prompts.

$$M = M_1; M_2; M_3; \ldots; M_i$$

The meta-prompt is applied to an input ($x$) to obtain the final prompt M{x} that conditions the LLM to generate the second positive caption ($P_2$) as $LLM(M\{P_1\}) \rightarrow P_2$. We find integrating different techniques into a larger meta-prompt ($M$) improves the generation quality, exploiting the benefits from each of them. Following [39], we prefix $M$ with a 'role-play' prompt ($M_1$) that conditions the LLM to the high-level task of data generation and simulates the role of a 'Data Generating AI.' We utilize 'Rules Prompting' [99] to enhance LLMs' faithfulness in instruction following using explicit and itemized rules: in particular, we further condition M using a 'rules' prompt ($M_2$) that describes three rules to ensure the consistency of the generated caption ($P_2$). Expanding on the prompting methodology from [29], we also append few-shot 'demonstrations' ($M_3$) that include additional reasoning and the generated caption. The 'role-play' ($M_1$) prompt is described in Figure 2. The 'rules' ($M_2$) and 'demonstrations' ($M_3$) sub-prompts are elaborated in Figure 3 of Appendix C. During our initial testing, we noticed the following systematically recurring errors: (i) LLM generated caption is *identical* to the input caption; (ii) parsing failure due to superfluous outputs. Consequently, we incorporate the following safeguards in the generation pipeline as shown in Algorithm 1: 1) If the generated caption is found to be identical to the input caption (using automatic tools), we use a meta-prompt that includes the complete context with basic instructions to regenerate $P_2$; 2) We detect superfluous outputs based on word overlap and then discard any generation that does not meet the minimum threshold.

**Automated and human validation of generated captions:** Since the above safeguards do not ensure that the generated caption is *always* semantically equivalent to the input caption, we require additional steps to ensure semantic equivalence. We notice subtle differences in generated caption that break the semantic equivalence and this also highlights the limitations of semantic understanding in LLMs. Despite such limitations, prior works [63, 13] have demonstrated that LLM agents — e.g., LLMs instantiated with different prompts — can interact with each other to solve complex tasks. We build on this to reduce human effort and optimize labelling costs using automatic validation of generated captions similar to [102, 99]. We employ a validator LLM agent, that is responsible for validating the semantic consistency of the generated caption with the original caption and signalling the Generator agent to retry as needed. Here, we refer to the LLM agents as two instances of the same LLM conditioned on different meta-prompts.

We define a validation meta-prompt (V) that uses $P_1$ and $P_2$ from the previous step to form a validation input to the validator LLM. Similar to the meta-prompt (M), the validation prompt (V) consists of a sequence of sub-prompts, including the 'Validation instruction' ($V_1$) prompt and the 'Validation demonstration' ($V_2$). The validator LLM is conditioned using V to generate a boolean value indicating the caption's validity,

$LLM(V\{P_1, P_2\}) \rightarrow \{True, False\}$, where $V = V_1; V_2$. We use the boolean output value to trigger a regeneration step that starts the generation pipeline again as described in Algorithm 2. The prompt for employing validator LLM is detailed in Figure 4 in Appendix C.3. In our automatic validation, we assume that the image is captioned correctly and do not consider the image when ensuring semantic correctness.

To ensure the quality of the SUGARCREPE++ dataset, we conducted human validation with two experts to correct the errors in the positive sentences ($P_2$) generated by LLMs (Appendix C.5 provides a list of common errors generated by LLMs) and any disagreements between the expert annotators were mutually resolved through inter-annotator discussion. These human annotators also assessed the validity of caption triplets ($P_1$, $P_2$, $N$) and paired images ($I$) as a data point. The final statistics of the

Table 1: SUGARCREPE++ consists of 4757 examples with the following distribution of sample sizes

| Swap Object | Swap Attribute | Replace Object | Replace Attribute | Replace relation | **Total** |
|---|---|---|---|---|---|
| 245 | 666 | 1652 | 788 | 1406 | 4757 |

SUGARCREPE++ dataset after the human validation are described in Table 1. Additional statistics, including the categories from MS-COCO and the VERA or grammar scores of SUGARCREPE++ are detailed in Appendix I. We formulated a measure of syntactic and lexical similarity to analyze pairs of sentences in our dataset. See Appendix D for more details.

## 3 Benchmark on SUGARCREPE++ Dataset

**Experimental setup** We benchmark VLM performance on SUGARCREPE++ dataset under two different evaluation settings. (1) Multi-modal image-to-text (ITT) evaluation: both image and text are provided as inputs to evaluate VLMs in a multi-modal setting. (2) Uni-modal text-only (TOT) evaluation: only text is provided as input to evaluate the text encoders of VLMs in a unimodal setting.

Each sample in the SUGARCREPE++ dataset consists of an image $I$ and corresponding two positive captions ($P_1$ and $P_2$) and a negative caption (N). If $p(X|I)$ denotes the likelihood of caption $X$ for image $I$, we compute the ITT evaluation metric given $P_1$, $P_2$ and N as:

$$\text{ITT}_{hit} = \begin{cases} 1 & (p(P_1|I) > p(N|I)) \wedge (p(P_2|I) > p(N|I)) \\ 0 & \text{otherwise} \end{cases}$$

For a VLM model such as CLIP which relies upon embeddings, the log-likelihood $\log p$ is defined to be proportional to the cosine similarity between the respective embeddings. Similarly, the TOT evaluation is defined as:

$$\text{TOT}_{hit} = \begin{cases} 1 & (p(P_1|P_2) > p(N|P_2)) \wedge (p(P_2|P_1) > p(N|P_1)). \\ 0 & \text{otherwise} \end{cases}$$

As above, we use cosine-similarity for embedding-based models. We report the performance in terms of Accuracy (%), computed as the ratio of the number of hits to the total number of samples. we also perform a human evaluation to calculate the human performance on the benchmark. Human evaluation is performed by 4 graduate-level students where each person was provided with a randomly selected 150 samples (30 from each subset) and was asked to select the negative caption for each sample. In TOT setting, only the three captions were provided. In the ITT setting, image along with the captions were provided to the human evaluators. The average human performance is reported in terms of accuracy (%).

### 3.1 Evaluation of VLMs on SUGARCREPE++

We consider a variety of VLMs for evaluation using SUGARCREPE++: 1) Models trained with a contrastive learning objective such as CLIP [66], RoBERTa-ViT-B/32 [76], ALIGN [35] and ALIP [95]. 2) Models trained by combining multiple objective functions, such as FLAVA [78], ALBEF [43], BLIP [44] and BLIP-2 [45]. 3) Models with a unified encoder for text and images, such as ViLT [38], and multi-lingual distilled models like AltCLIP [16]; 4) Models that align text with corresponding visual concepts in the image, such as SegCLIP [54], and XVLM [98] - with two variants, XVLM-4M and XVLM-16M. We consider a wide array of VLMs that differ in terms of model architecture, total number of parameters, embedding dimension and pre-training objectives to measure the effect of various training choices on the model's semantic understanding capabilities. For further model details, refer to Appendix E.1.

**Performance of VLMs on SUGARCREPE++ is strongly influenced by the type of hard negative.** The performance of VLMs on different subsets of SUGARCREPE++ are provided in Table 2. Swap type hard negatives, which are generated by swapping either Objects or attributes, pose a significant challenge to VLMs, as most achieve very low performance. Failing in examples with simple reordering of words, as in swap subset, highlights a key limitation of VLMs in understanding the structure of the input text. For replace-type hard negatives (generated by replacing objects, attributes,

or relations), VLMs are comparatively better at discerning the negative from the positive caption when the object is replaced. VLMs can also somewhat discern hard negatives from positives when attributes are replaced. However, they struggle when the relation between objects is replaced. Additionally, large performance gaps (ranging from 10% to 50%) between the best models and human performance across most subsets signifies opportunity for improvement in VLMs' semantic understanding abilities.

**Pre-training data size and objective functions affect VLM performance.** Table 2 shows that the models trained with multiple objective functions, particularly FLAVA and BLIP, perform better on SUGARCREPE++ compared to models trained using contrastive loss function alone. This indicates that the contrastive learning objective alone may not be sufficient for VLMs to effectively learn the semantic relations between text and images. Furthermore, models pre-trained with smaller datasets, such as ALIP, ALBEF and XVLM-4M, have lower performance compared to other models. Interestingly, these observations are consistent across all subsets of SUGARCREPE++.

**Text encoders bottleneck VLM performance on SUGARCREPE++.** All VLMs perform significantly better on the ITT task compared to the TOT task on SUGARCREPE++ (see Table 2). This shows that there is a higher ambiguity in identifying the semantic and lexical alterations using only the text embeddings (TOT) compared to the case of comparing the text embeddings with the image embeddings (ITT). Moreover, the text encoders of VLMs perform inferior to the text encoders of ULMs (see Table 7). This is in agreement with the findings in [37, 15]. Additionally, FLAVA, the best performaning model on most of the subsets also achieves a good performance in TOT setting, further signifying the importance of a strong text encoder in achieving better performance.

**Fine-tuning VLMs for image-text retrieval improves performance with opportunity for further improvements.** Table 3 provides the performance of VLMs (ViLT and XVLM-16M) fine-tuned for the task of image-to-text retrieval (ITR). While we observe performance improvements on SUGARCREPE++, VLMs still face significant challenges in discerning negative captions from positive ones, particularly for the Swap object and Replace relation subsets. Moreover, there remains a substantial gap between VLM performance and human-level performance. This indicates that VLMs capable of matching images to corresponding captions do not necessarily learn the intricate details regarding semantic information and lexical variations in the text.

Table 2: Comparison of VLMs performance on SUGARCREPE++. Performance reported in terms of Accuracy (%). Overall best values are in bold, and group-level best values are underlined.

| Model | Swap Object | | Swap Attribute | | Replace Object | | Replace Attribute | | Replace Relation | |
|---|---|---|---|---|---|---|---|---|---|---|
| | ITT | TOT | ITT | TOT | ITT | TOT | ITT | TOT | ITT | TOT |
| Human | 100.00 | 96.67 | 96.67 | 93.3 | 100.00 | 95.00 | 100.00 | 98.33 | 100.00 | 96.67 |
| CLIP [66] | 45.18 | 19.74 | 45.21 | 33.03 | 86.80 | 83.72 | 65.61 | 59.14 | 56.26 | 38.62 |
| RoBERTa-ViT-B/32 [76] | 44.30 | 29.39 | 56.32 | 52.66 | 89.04 | 94.55 | 74.49 | 80.46 | 59.39 | 57.75 |
| ALIGN [35] | 41.23 | 25.43 | 51.90 | 41.40 | 90.19 | 84.62 | 69.92 | 69.04 | 51.71 | 45.23 |
| ALIP [95] | 36.84 | 20.18 | 46.12 | 28.77 | 71.49 | 50.06 | 54.95 | 34.52 | 47.80 | 23.47 |
| FLAVA [78] | **54.39** | **45.18** | 59.21 | **57.84** | 89.59 | 94.43 | 73.35 | 72.46 | **60.10** | **57.97** |
| ALBEF [43] | 28.94 | 10.09 | 36.83 | 18.87 | 76.27 | 55.57 | 56.35 | 30.33 | 47.80 | 30.65 |
| BLIP [44] | 47.37 | 31.14 | **60.58** | 52.97 | **92.62** | 89.04 | 72.08 | 75.13 | 56.76 | 57.47 |
| BLIP2 [45] | 35.09 | 21.49 | 37.60 | 29.98 | 89.41 | 72.58 | 62.82 | 64.47 | 53.27 | 43.47 |
| ViLT [38] | 35.23 | – | 52.20 | – | 91.10 | – | 55.33 | – | 37.48 | – |
| AltCLIP [16] | 42.54 | 25.43 | 45.81 | 35.77 | 92.61 | 93.46 | 71.06 | 74.62 | 57.25 | 56.69 |
| SegCLIP [54] | 45.61 | 25.44 | 46.12 | 40.64 | 85.90 | **95.16** | 62.69 | 67.89 | 54.84 | 41.96 |
| XVLM-4M [98] | 31.14 | 10.96 | 36.52 | 19.48 | 79.42 | 67.07 | 59.39 | 40.74 | 46.23 | 29.23 |
| XVLM-16M [98] | 34.21 | 18.86 | 40.33 | 31.05 | 90.81 | 92.07 | 68.02 | 70.43 | 57.47 | 47.87 |

Table 3: Evaluation of VLMs fine-tuned for image-text retrieval task. Performance reported in terms of Accuracy (%). ITR dataset is the dataset used to fine-tune the model for image-to-text retrieval.

| Model | ITR Dataset | Swap Object | | Swap Attribute | | Replace Object | | Replace Attribute | | Replace Relation | |
|---|---|---|---|---|---|---|---|---|---|---|---|
| | | ITT | TOT | ITT | TOT | ITT | TOT | ITT | TOT | ITT | TOT |
| ViLT [38] | – | 35.23 | – | 52.20 | – | 91.10 | – | 55.33 | – | 37.48 | – |
| XVLM-16M [98] | – | 34.21 | 18.86 | 40.33 | 31.05 | 90.81 | 92.07 | 68.02 | 70.43 | 57.47 | 47.87 |
| ViLT-ITR-COCO [38] | MS-COCO | **50.88** | – | **73.36** | – | 89.89 | – | 71.95 | – | 61.24 | – |
| XVLM-16M-COCO [98] | MS-COCO | 39.91 | 21.06 | 49.93 | **51.60** | 91.65 | **96.79** | 74.24 | **83.63** | 63.09 | **62.87** |
| XVLM-16M-Flickr [98] | Flickr | 45.61 | **21.49** | 50.53 | 44.44 | **91.71** | 96.01 | 74.24 | 81.59 | **64.01** | 59.89 |

Table 4: Performance of the methods for improving compositionality of VLMs on SUGARCREPE++. Performance reported in terms of Accuracy (%). Here, performance of CLIP is the baseline.

| Model | Swap Object | | Swap Attribute | | Replace Object | | Replace Attribute | | Replace Relation | |
|---|---|---|---|---|---|---|---|---|---|---|
| | ITT | TOT | ITT | TOT | ITT | TOT | ITT | TOT | ITT | TOT |
| Human | 100.00 | 96.67 | 96.67 | 93.3 | 100.00 | 97.00 | 100.00 | 98.33 | 100.00 | 96.67 |
| CLIP [66] | 45.18 | 19.74 | 45.21 | 33.03 | 86.80 | 83.72 | 65.61 | 59.14 | **56.26** | 38.62 |
| NegCLIP [97] | **55.25** | **34.65** | **57.99** | **56.47** | **89.53** | **94.55** | **69.41** | **76.27** | 52.27 | **51.57** |
| CLIP-SVLC [21] | 42.98 | 18.86 | 48.40 | 34.56 | 80.93 | 91.56 | 56.98 | 66.88 | 47.30 | 51.28 |
| BLIP-SGVL [26] | 13.16 | – | 38.81 | – | 53.75 | – | 34.39 | – | 30.65 | – |
| CyCLIP [23] | 37.72 | 13.60 | 34.40 | 18.72 | 70.28 | 78.29 | 49.87 | 49.12 | 40.41 | 29.87 |

**Compositionality enhancing methods improve performance on SUGARCREPE++ by strengthening the VLM text encoder.** We evaluate recent methods proposed to improve compositionality of VLMs, including NegCLIP [97], SVLC [21], CyCLIP [23], and BLIP-SGVL [26]. As shown in Table 4, methods that improve compositionality of CLIP such as NegCLIP and CLIP-SVLC also achieve better performance on SUGARCREPE++ compared to CLIP, underscoring the importance of compositionality as a critical component for understanding semantic and lexical differences. Interestingly, the text encoders of models with improved compositionality (NegCLIP and CLIP-SVLC) perform significantly better than the text encoder of CLIP in the TOT setting. Improved text encoders, in turn, lead to improvements in ITT. On the other hand, methods such as BLIP-SGVL and CyCLIP, which do not use explicit techniques to strengthen the text encoders, show degradation in performance on SUGARCREPE++. This further highlights the importance of the text encoder in achieving better performance.

Table 5: Evaluation of several variants of CLIP on SUGARCREPE++. Performance reported in terms of Accuracy (%). Best performances in bold. RoB refers to RoBERTa.

| Model | #Model Params | Pre-train Data Size | Swap Object | | Swap Attribute | | Replace Object | | Replace Attribute | | Replace Relation | |
|---|---|---|---|---|---|---|---|---|---|---|---|---|
| | | | ITT | TOT | ITT | TOT | ITT | TOT | ITT | TOT | ITT | TOT |
| CLIP [66] | 151M | 400M | 45.18 | 19.74 | 45.21 | 33.03 | 86.80 | 83.72 | 65.61 | 59.14 | 56.26 | 38.62 |
| RN50×4 [66] | 178M | 400M | **46.93** | 21.49 | 46.42 | 30.59 | 87.77 | 80.87 | 67.51 | 53.93 | 53.91 | 38.55 |
| RN50×64 [66] | 623M | 400M | 44.74 | 16.67 | 45.36 | 31.51 | 90.79 | 73.31 | 64.47 | 48.61 | 54.27 | 38.12 |
| RoB-ViT-B/32 [76] | 212M | 2000M | 44.30 | 29.39 | 56.32 | 52.66 | 89.04 | 94.55 | 74.49 | 80.46 | 59.39 | 57.75 |
| ViT-H/14 [76] | 986M | 2000M | 43.42 | 27.63 | 54.19 | 50.69 | 93.71 | 90.43 | 71.06 | 73.98 | 56.62 | 51.92 |
| ViT-bigG/14 [76] | 2540M | 2000M | 45.61 | 29.82 | **57.38** | 52.05 | **94.13** | 90.44 | **76.41** | 72.84 | **59.45** | 53.49 |
| XLM-RoB-ViT-B/32 [76] | 366M | 5000M | 42.55 | **30.26** | 55.25 | **55.56** | 89.41 | **95.34** | 72.97 | **80.96** | 55.48 | **57.82** |

**Larger pre-training data and model size improve CLIP's performance.** We evaluated variants of CLIP that differ in model architecture and size, as well as pre-training data size, on SUGARCREPE++ (see Table 5). For CLIP variants pre-trained on a dataset of 400 million image-text pairs, smaller models (CLIP and RN50×4) performed better than larger models (RN50×64). Interestingly, larger models (ViT-bigG/14) performed better than smaller models when the pre-training data was increased to 2 billion samples. Moreover, the text encoders also performed better when the pre-training data was increased to 2 billion image-text pairs.

**Fine-tuning on SUGARCREPE++ does not necessarily improve semantic understanding of VLMs.** Table 13 (see Appendix E.3) presents the fine-tuning results.hl Experimental results show that while there are performance improvements on some subsets, fine-tuning can lead to performance degradation on others, which may be attributed to catastrophic forgetting [55]. To address this issue, we used LoRA [30]. Results demonstrate that LoRA improved performance across all subsets.

**Comparison of performance between SUGARCREPE and SUGARCREPE++ reveals significant differences.** Table 6 compares the performance of VLMs between SUGARCREPE and SUGARCREPE++. Since a direct comparison of the models' absolute performance on SUGARCREPE and SUGARCREPE++ is not possible, we assess their relative rankings using CLIP as a baseline. Notably, we observe significant differences in performance trends across the models. For instance, ALBEF and XVLM which achieve better performance on SUGARCREPE show substantial degradation in performance on SUGARCREPE++. On the other side, there are models such as BLIP and NegCLIP that show improvements on both SUGARCREPE and SUGARCREPE++. Interestingly for the replace relation subset, NegCLIP achieves better performance on SUGARCREPE but shows degradation in performance on SUGARCREPE++. These results emphasize the importance of our dataset in

evaluating the models in terms of their semantic and lexical understanding, which is not evident by evaluating on compositionality datasets such as SUGARCREPE.

Table 6: Comparing the performance of VLMs on SUGARCREPE (SC) and SUGARCREPE++ (SC++) for the ITT task. ↑ and ↓ show increases and decreases in performance with the corresponding CLIP performance as the baseline. Expanded version of the Table is provided in Appendix E.5.

| Model | Swap Object | | Swap Attribute | | Replace Object | | Replace Attribute | | Replace Relation | |
|---|---|---|---|---|---|---|---|---|---|---|
| | SC | SC++ | SC | SC++ | SC | SC++ | SC | SC++ | SC | SC++ |
| CLIP [66] | 59.21 | 45.18 | 64.99 | 45.21 | 90.86 | 86.8 | 80.33 | 65.61 | 70.48 | 56.26 |
| ALBEF [43] | 63.16 (↑) | 28.94(↓) | 69.25(↑) | 36.83(↓) | 93.04(↑) | 76.27(↓) | 84.65(↑) | 56.35(↓) | **77.60**(↑) | 47.80(↓) |
| XVLM [98] | 64.91(↑) | 31.14(↓) | 73.97(↑) | 36.52(↓) | 95.22(↑) | 79.42(↓) | **87.69**(↑) | 59.39(↓) | 77.45(↑) | 46.23(↓) |
| BLIP [44] | 66.22(↑) | 47.37(↑) | 76.25(↑) | **60.58**(↑) | **96.55**(↑) | **92.62**(↑) | 81.98(↑) | **72.08**(↑) | 68.35(↓) | **56.76**(↑) |
| NegCLIP [97] | **75.44**(↑) | **55.25**(↑) | **76.87**(↑) | 57.99(↑) | 93.88(↑) | 89.53(↑) | 87.18(↑) | 69.41(↑) | 74.47(↑) | 52.27(↓) |

## 3.2 Evaluation of ULMs on SUGARCREPE++

**ULMs show promise over VLMs in text-only task of SUGARCREPE++.** We evaluate a comprehensive list of unimodal language models (ULMs) to determine the semantic and lexical sensitivity of text-only models. We sample ULMs covering various model sizes, architectures, and training objectives. Recent state-of-the-art small-sized models include MiniLM [70], GTE [47] and BGE [93]. From the results presented in Table 7, we observe trends similar to the VLM's performance in TOT task but more importantly, we observe ULMs achieve significant improvements on average as compared to VLMs. Nevertheless, we notice that some of these performances are far below human performance — for instance, we observe a performance gap of ≈40% in swap object and swap attribute subsets. Comprehensive results are available in Appendix F.

Table 7: Comparison of SOTA unimodal language models (ULMs) on SUGARCREPE++. We report the TOT accuracy (%), and group the results row-wise based on the model size as reflected by the parameter count. We include the number of parameters in text encoders relative to BERT-base, i.e., 109.5 Million parameters. Overall, best values are in bold, and group-level best values are underlined. We report the average across different subsets as an additional column. Refer to Table 16 for additional results.

| Model | #Params (BERT Scale) | Swap Object | Swap Attribute | Replace Object | Replace Attribute | Replace Relation | Average |
|---|---|---|---|---|---|---|---|
| Human | | 96.67 | 93.3 | 97.00 | 98.33 | 96.67 | 96.40 |
| BGE-small-en-v1.5 [93] | 0.3 | 15.51 | 24.02 | 94.19 | 75.00 | 75.53 | 56.85 |
| All-MiniLM-L12-v2 [91] | 0.3 | 18.78 | 25.38 | 95.22 | 73.86 | 70.41 | 56.73 |
| Angle-BERT-base [46] | 1 | 25.71 | 33.63 | 92.07 | 78.43 | 75.32 | 61.03 |
| BGE-base-en-v1.5 [93] | 1 | 17.14 | 25.23 | 93.83 | 78.55 | 76.10 | 58.17 |
| UAE-Large-v1 [46] | 3 | 40.41 | 41.44 | **96.85** | 76.14 | 75.82 | 66.13 |
| All-RoBERTa-large-v1 [70] | 3.1 | 42.04 | 45.20 | 94.61 | 74.75 | 74.96 | 66.31 |
| Sentence-T5-xl [62] | 11.3 | **47.35** | **49.25** | 90.98 | 75.38 | 75.32 | 67.66 |
| Angle-Llama-7b-nli-v2 [46] | 62.3 | 37.96 | 45.80 | 95.22 | 84.39 | **81.44** | **68.96** |
| E5-Mistral-7b-instruct [89, 88] | 64.9 | 33.47 | 37.84 | 96.67 | **87.06** | 80.51 | 67.11 |

## 3.3 Evaluation on GPT-4o and Other Generative VLMs on SUGARCREPE++

We evaluated generative VLMs such as BLIP, BakLLaVA [53] and GPT-4o using SUGARCREPE++ dataset in a prompt-based format. We prompted the generative VLMs with the following semantically identical prompts, where N, $P_1$, $P_2$ refer to the negative (N) and the two positive captions ($P_1$ and $P_2$) corresponding to the image, respectively.

**Prompt - 1** Do any of the following captions not match the image? (1) <N>; (2)<$P_1$>"; (3) <$P_2$>; provide output as (1), (2), (3) or none

**Prompt - 2** Do any of these captions fail to correspond with the image? (1) <N>; (2) <$P_1$>; (3) <$P_2$>; provide output as (1), (2), (3) or none

**Prompt - 3** Do any of these captions fail to correspond with the image? (1) <N>; (2) <$P_1$>; (3) <$P_2$>; provide output as (1), (2) or (3)

First two prompts (Prompt-1 and Prompt-2) are paraphrases of each other and are 4-class problems (Output to be (1), (2), (3) or none). Whereas Prompt-3 is a 3-class problem i.e., outputs to be (1), (2) or (3).

Table 8 provides GPT-4o's performance on SUGARCREPE++ across the three different prompts. Significant differences between Prompt-1 and Prompt-2, which are paraphrases, highlight the model's sensitivity to prompt structure. GPT-4o struggled to identify negative captions when given four options ((1), (2), (3), or none) but improved when limited to three ((1), (2), or (3)). The model performed best when negative captions were created by replacing an object or attribute in the positive caption but struggled when objects or relations were swapped. While GPT-4o's performance on "replace object" and "replace attribute" tasks neared human levels, it fell short significantly in "swap object," "swap attribute," and "replace relation" cases. Prompt-based evaluation of BLIP and BakLLaVA are provided in Appendix G.1.

Table 8: Prompt-based evaluation of GPT-4o on SUGARCREPE++. We provide both image and a prompt to the GPT-4o and receive the output from GPT-4o. Based on the response, we compute the performance i.e., it is a hit if the model outputs (1) else a miss. Performance is reported in terms of Accuracy (%), where accuracy is computed as the ratio of the #hits/(#hits + #misses).

|          | Swap Object | Swap Attribute | Replace Object | Replace Attribute | Replace Relation |
|----------|-------------|----------------|----------------|-------------------|------------------|
| Human    | 100.00      | 96.67          | 100.00         | 100.00            | 100.00           |
| Prompt-1 | 46.93       | 73.36          | 91.64          | 87.94             | 69.06            |
| Prompt-2 | 48.25       | 75.04          | 90.82          | 84.90             | 71.19            |
| Prompt-3 | **67.61**   | **85.82**      | **96.25**      | **93.27**         | **84.13**        |

**Inference techniques can influence the performance of generative VLMs on SUGARCREPE++.** We evaluated generative VLMs on SUGARCREPE++ (see Appendix G.2) using recent approaches such as VGPTScore [50] and VQAScore [51, 42]. We find that despite using significantly larger models (3B-11B parameters), VGPTScore performs comparably with several discriminative VLMs previously considered in our paper (e.g., Table 2). Interestingly, using the same generative VLMs, VQAScore achieves significant performance improvements on SUGARCREPE++ as compared to VGPTScore. However, this is still below human performance signifying opportunity for further improvement. Based on several experiments with generative VLMs, we can conclude that the performance of generative VLMs ultimately depends on the inference-technique (e.g., VGPTScore/VQAScore/Prompting-styles) and with the right inference-technique, generative-VLMs may outperform discriminative models.

## 4 Conclusion

We introduce SUGARCREPE++, a dataset for evaluating the ability of language models, including both vision-language models (VLMs) and unimodal language models (ULMs), to understand their sensitivity to semantic and lexical alterations in text. Our dataset supports evaluations in both image-to-text (ITT) and text-to-text (TOT) settings. We evaluated a comprehensive list of VLMs and ULMs to highlight a fundamental limitation with these language models in understanding semantic and lexical alterations. The key findings from our evaluation are: (1) There is a significant performance gap between VLMs and human-level performance signifying huge scope for improvement in VLMs. (2) All VLMs exhibit difficulty in comprehending semantic and lexical alterations, especially when these alterations involve swapping attributes or objects, or replacing relations. (3) Similarly, state-of-the-art (SOTA) ULMs lack a robust understanding of lexical composition and consistently fail to separate semantics from lexical forms. (4) While increasing pre-training data, model size, and improving compositionality enhance performance on SUGARCREPE++ dataset, these models still fall considerably short of human performance. These insights underscore the critical need for advancements to close the performance gap between models and human understanding. Our SUGARCREPE++ dataset serves as a valuable tool for driving future research in this area.

## Acknowledgments

We acknowledge the support provided by the Faculty of Computer Science, Dalhousie University. Resources used in preparing this research were provided, in part, by the support of the Natural Sciences and Engineering Research Council of Canada (NSERC), the Province of Ontario, the Government of Canada through Canadian Institute for Advanced Research (CIFAR), ACENET (acenet.ca), the Digital Research Alliance of Canada (alliancecan.ca), Canada Foundation of Innovation and companies sponsoring the Vector Institute www.vectorinstitute.ai/#partners. We are grateful to the anonymous reviewers for their insightful feedback and active engagement throughout the rebuttal process, which was instrumental in strengthening our paper.

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

# Appendix

## Table of Contents

# A  Statements of Limitations, Impact & Contribution

## A.1  Limitation and Future Work

In this paper, we evaluate a large set of VLMs on our proposed SUGARCREPE++ dataset. We provide observations based on the results even though it is difficult to provide comparison between models as the models differ in terms of model architecture, pre-training data content and size, pre-training objectives, etc. Unless we can train train separate models for each setting where one of the parameters is kept constant, it is difficult to draw definitive conclusions. Providing guidelines to improve the performance of VLMs on SUGARCREPE++ dataset depends on the observations we draw from the above mentioned analysis. Thus, in this paper, we limit to identifying one of the major issue with current VLMs, which can probe further research in this direction. Moreover, our dataset can be used to evaluate models to access their ability to discern lexical alterations from semantic alterations. We followed a simple procedure for human evaluation of SUGARCREPE++ tasks in this paper, which might not give human performance that can be directly compared to the embedding-based performance of VLMs and ULMs. A better and more rigorous approach to evaluate the ability of humans in understanding the affect of altering semantic and lexical content in the input text can be proposed in the future work.

## A.2  Impact Statement

This paper presents work whose goal is to advance the field of Machine Learning in general and Language Models research in particular. We discuss several limitations of language models related to the separation of semantics of an input text from its syntactic and lexical form. In order to build trust-worthy Language Models, it is important to establish that the language models emphasize semantics contained in a sentence rather than the lexical form and syntactic style of the sentence. Our evaluation provides evidence of this problem through two curated datasets and can potentially be impactful for evaluating newer language models and/or inspiring novel solutions to this problem. There are many other potential societal consequences of our work, none of which we feel must be specifically highlighted here.

## A.3  Contribution Statement

SHD, AJ and CSS are the student contributors in this project and the following summarizes their contributions to this paper: SHD led the research efforts in developing a novel contribution and wrote the first draft of the paper as well as the rebuttals. SHD benchmarked the VLMs and created a first draft of the visualisations. AJ worked on the research and development of a pipeline for automatic paraphrase generation and validation before the human verification step. AJ worked on benchmarking the ULMs, created the visualisations in the paper, led the efforts in creating the final shareable dataset and drafted the corresponding sections. CSS proposed the 3-way testing approach for evaluating semantic consistency in language embeddings and helped with finalizing the paper draft and rebuttals.

# B  Related Work

VLMs and ULMs have achieved impressive results on a range of vision and language downstream tasks. These state-of-the-art VLMs and ULMs serve as foundation models for both multimodal applications, like image captioning [45], semantic segmentation of images [18, 48], text-to-image generation [67, 68, 73], and unimodal applications, like clustering [89, 88], reranking [93], and retrieval [46]. Their emergence as foundation models has motivated recent research to evaluate the strengths and weaknesses of these models. We summarize the findings from common benchmarks of VLMs and ULMs below.

**Findings from the existing benchmarks for VLMs:**  Thrush et al. [83] evaluate VLMs through an image-text retrieval task and find that SOTA VLMs struggle to distinguish between texts containing the same words but ordered differently. Similarly, Yuksekgonul et al. [97] evaluate VLMs in terms of their abilities to form object-attribute associations and highlight shortcomings of VLMs. Other studies with similar conclusions include [100], [69] and [87]. Recent works have introduced benchmarks to evaluate different abilities of VLMs such as counter-intuitive reasoning [103], visual question answering [94], conceptual understanding [75], visio-linguistic reasoning [17], visual-spatial reasoning [52] and compositionality [83]. Kamath et al. [37] demonstrate challenges in decoding salient aspects of input text encoded with CLIP and draw connections to the lack of compositionality in CLIP text embeddings.

The task of evaluating compositionality in VLMs is the nearest neighbor to our work. Several datasets have been introduced to evaluate the compositionality of VLMs [52, 29, 83, 100, 97, 56, 69, 87, 74]. Most of the existing compositionality benchmarks formulate the evaluation task as image-text retrieval. Winoground [83] is one of the earliest benchmarks to report the lack of compositional understanding in VLMs. Latest benchmarks encompassing different aspects of compositionality include VL-CheckList [100], CREPE [56], Cola [69], and ARO [97]. Some benchmarks like Winoground have challenges beyond compositionality that include additional visual and textual reasoning [19].

**Findings from the existing benchmarks for ULMs:**  In the context of ULM text encoders, paraphrasing is the closest to our paper. Paraphrasing is a well-studied problem in NLP. Several previous studies analyzed the ability of the language models to recognize paraphrasing in text. The Microsoft Research Paraphrase Corpus (MRPC) [20] and Quora Question Pairs (QQP) [34] are popular paraphrasing datasets (text-only without images) that are part of the GLUE (General Language Understanding Evaluation) [86] benchmark. The Semantic Textual Similarity (STS) benchmark [12] build from the STS shared tasks [2–6] have pairs of text snippets with scores indicating the degree of semantic equivalence between them.

**Shortcomings of existing benchmarks:**  Alper et al. [7] find that the CLIP text encoder outperforms the ULMs in tasks that require implicit visual reasoning, while Chen et al. [15] find that ULMs perform better in terms of general language understanding. Most VLM benchmarks are generated using rule-based algorithms [56, 97] and consist of only a pair of sentences (either semantically similar or dissimilar sentences). These similar pairs might not have strong semantic similarities, and the dissimilar pairs can have significant lexical differences, which does not represent a strict setting of evaluation. Moreover, we must finetune or linearly probe [52] these models to evaluate LLM encoders using these datasets, which can require significant resources. None of the existing benchmarks systematically evaluates the resilience of model embeddings in the presence of lexical distractors [31, 82], i.e., lexically similar but semantically different negative inputs.

# C SUGARCREPE++ Benchmark Generation

## C.1 Dataset Guidelines

The main guidelines followed to create the benchmarks are:

- The lexical changes allowed for creation of the three captions include, replacing words with synonyms and antonyms, reordering the words, etc. These lexical changes do not include adding more details about the image in the caption.
- Due to the lexical alterations, the three captions should not consist of any nonsensical and non-fluent errors.
- The three captions should be generated such that they do not need any visual, logical or commonsense reasoning to distinguish the semantically similar captions ($P_1$ and $P_2$) from the semantically different caption (N) i.e., given only three captions without image, one should be able to distinguish $P_1$, $P_2$ from N.

## C.2 Prompt for SUGARCREPE++

Figure 3 shows the prompts used to condition the generation of $P_2$ using LLM, given $P_1$. Here, we use instruct fine-tuned Mistral 7B [36] model to generate $P_2$.

---

**Rules Instruction:** Given an input sentence describing an image caption, follow these steps:

1. Rephrase each provided sentence, focusing on preserving the original spatial relationship.
2. Pay careful attention to the positioning of objects or entities in relation to one another.
3. Ensure that the meaning remains consistent and that both the original and paraphrased sentences maintain logical coherence and grammatical correctness.

**Demonstration:** For example,

**Input:** Cat is under the table.
**Paraphrase Idea:** Rephrase the sentence to convey that the table is positioned above the cat.
**Paraphrased:** The table is above the cat.

**Input:** The plane flies below the bright white clouds.
**Paraphrase Idea:** Ensure the spatial context is maintained by stating that the bright white clouds are situated above the flying plane.
**Paraphrased:** The plane flies below the bright white clouds.

**Input:** The third balcony is below the fourth balcony from the bottom.
**Paraphrase Idea:** Emphasize the consistent spatial arrangement while indicating that the fourth balcony is positioned above the third balcony from the bottom.
**Paraphrased:** The fourth balcony is above the third balcony from the bottom.

*Remember to keep the meaning intact, and both the original and paraphrased sentences should be logically coherent and grammatically correct.*

**Input:** *[Original caption goes here]*
**Paraphrase Idea:** Focus on replicating the spatial arrangement while maintaining the original meaning of the sentence, correct grammar, same meaning.
**Paraphrased:** *[Your paraphrased sentence goes here]*

---

Figure 3: Rules and demonstration sub-prompts used to condition the generator LLM.

## C.3 Validation Prompt for SUGARCREPE++

Figure 4 shows the comprehensive prompt used to validate the samples generated by priming the LLM. The outputs obtained from this prompt are further validated by a human expert. This reduces the manual effort required to create the SUGARCREPE++ dataset.

---

**Instruction:** Given a pair of captions, you must check if the second caption is semantically consistent with the first caption. If the second caption is consistent, output the second caption as is; otherwise, rephrase the second caption to be consistent with the first sentence. We are interested in preserving the spatial consistency and spatial relationship of the objects with each other.

**Demonstrations:** examples,

**Caption 1:** A guy holding a skateboard is speaking into a microphone.
**Caption 2:** The guy holding the microphone is speaking into the skateboard.
**isConsistent**: No, you cannot speak into a skateboard.
**newCaption**: The guy is speaking into the microphone while holding a skateboard.

**Caption 1:** A family are playing frisbee on the beach.
**Caption 2:** The frisbee is being played on the beach by a family.
**isConsistent**: Yes, caption 2 is consistent as it is the same caption written in passive voice. new caption is the same as caption 2.
**newCaption**: A family are playing frisbee on the beach.

**caption 1:** A stop sign vandalized with an "eating animals" sticker below the word "stop."
**caption 2:** The stop sign is below an "eating animals" sticker.
**isConsistent**: The stop cannot be below and above the sticker at the same time.
**newCaption**: The word "stop" sign is above an "eating animals" sticker.

**caption 1:** There is a phone on top of a calculator.
**caption 2:** A calculator lies beneath the phone.
**isConsistent**: Yes, the sentences are semantically equivalent. new caption is same as caption 2.
**newCaption**: A calculator lies beneath the phone.

Now the same for the below caption only.
**caption 1:** *[Original caption goes here]*
**caption 2:** *[Generated caption goes here]*
**isConsistent**: *[Output Here]*

---

Figure 4: Validation Meta-prompt used to validate the consistency of the generated caption and original caption. We use the *isConsistent* output to signal the regeneration of a semantically inconsistent caption.

## C.4 Negative Caption Description

We use five negative caption categories from the SUGARCREPE dataset [29]. Hsieh et al. [29] describe the generation procedure for these hard negative captions in Appendix C2 of their paper. Below, we provide a brief description of these negative caption categories.

(a) **Replace-Object:** Replaces a noun in the scene description with a new noun, creating a distinctly different scene while keeping the sentence fluent and logical.
(b) **Replace-Attribute:** Replaces an adjective describing an object with a new adjective to alter the scene meaningfully, ensuring grammatical correctness.
(c) **Replace-Relation:** Changes a spatial or action-based relationship between objects, resulting in a different scene description that is fluent and coherent.

(d) **Swap-Object:** Swaps two noun phrases in the sentence to form a new scene description that is grammatically correct and logically distinct from the original.

(e) **Swap-Attribute:** Exchanges two adjectives describing different objects to create a sentence that describes a new scene.

### C.5 Incorrect Generation Artifacts by LLMs

As shown in Figure 5, We observe various generation artifacts in the positive sentences ($P_2$) generated by LLMs that were corrected during the human validation stage.

---

**Negligible Changes**

| | |
|---|---|
| **Original Caption:** | A person in a green shirt is taking a picture with his cell phone. |
| **Generated Caption:** | A person in a green shirt is taking a picture with his cell phone. |
| **Manual Correction:** | An individual wearing a green shirt is using his cell phone to take a photo. |
| **Original Caption:** | Two small birds walking on a sidewalk next to patches of grass. |
| **Generated Caption:** | Two small birds are walking on a sidewalk next to patches of grass. |
| **Manual Correction:** | Couple of little birds are strolling on a sidewalk alongside patches of grass. |

---

**Hallucination**

| | |
|---|---|
| **Original Caption:** | The woman is enjoying playing a Wii interactive video game. |
| **Generated Caption:** | The woman is playing a Wii interactive video game while sitting on a couch. |
| **Manual Correction:** | The woman is playing a Wii interactive video game and she is enjoying it. |
| **Original Caption:** | A black and white photo of a farmer standing next to horses. |
| **Generated Caption:** | A photo of a farmer standing next to horses, with the horses positioned to the right of the farmer. |
| **Manual Correction:** | A monochrome photograph featuring a farmer standing alongside horses. |

---

**Incorrect Spatial Setting**

| | |
|---|---|
| **Original Caption:** | Two bear cubs are playing on a log. |
| **Generated Caption:** | The log is positioned above the two bear cubs while they are playing on it. |
| **Manual Correction:** | The log is positioned under two bear cubs who are playing on it. |
| **Original Caption:** | A dog is laying with its water dish in front of it; |
| **Generated Caption:** | The dog is positioned in front of its water dish. |
| **Manual Correction:** | The water dish is positioned in front of the lying dog. |

---

**Altering Semantic Meaning**

| | |
|---|---|
| **Original Caption:** | A man using his laptop computer while a cat sits on his lap. |
| **Generated Caption:** | A man is seated with a laptop computer on his lap while a cat rests on top of him. |
| **Manual Correction:** | A man with a cat sitting on his lap is using his laptop. |
| **Original Caption:** | A man is drinking water from a red bottle. |
| **Generated Caption:** | The red bottle is positioned above the man as he drinks water from it. |
| **Manual Correction:** | A man is sipping water from a bottle that is red in color. |

---

**Superfluous outputs**

| | |
|---|---|
| **Original Caption:** | A camera is shown viewing 3 cupcakes, one of which is being touched |
| **Generated Caption:** | A camera is shown viewing 3 cupcakes, one of which is being touched. Both the original and paraphrased sentences convey the same meaning and maintain logical coherence and grammatical correctness. |
| **Manual Correction:** | Three cupcakes are visible in the camera's view, with one of them being touched. |
| **Original Caption:** | A room with a bed, a desk, and a television. |
| **Generated Caption:** | The room with a bed, a desk, and a television, arranged in the same manner as the original sentence. |
| **Manual Correction:** | The room contains a television, a bed and a desk. |

---

Figure 5: Examples of **common errors in the LLM generated positive sentences ($P_2$)**. We provide five of the most common types of errors. Manual correction refers to the corrected sentences after the human validation step. Expert human annotators carefully checked each output of the LLM and corrected the erroneous sentences to maintain grammatical and semantic equivalence with the original positive caption ($P_1$)

# D   Lexical, Syntactical and Semantic Properties

In the following, we provide definitions, describe a metric for measuring lexical/syntactic similarities and visualize the statistics of this metric applied to SUGARCREPE++ dataset.

(a)  *Lexical [58]*: of or relating to words or the vocabulary of a language as distinguished from its grammar and construction.
(b) *Syntax [60]*: the way in which linguistic elements (such as words) are put together to form constituents (such as phrases or clauses).
(c) *Semantic [59]*: of or relating to meaning in language.

Together, lexical and syntactic aspects of the sentence refer to the choice of words and writing style used to construct a sentence. The meaning expressed by the sentence is referred to as semantics. By varying the lexical and syntactic aspects, we can create different sentences that express the same meaning (i.e., semantically equivalent). Likewise, we can create sentences that are semantically non-equivalent but are very close in terms of their lexical and syntactic aspects. By extending SUGARCREPE, SUGARCREPE++ consists of 3 captions for each image ($P_1$, $P_2$ & N) where $P_1$ and $P_2$ are correct captions (see Fig. 1 of our paper) and are related as follows:

$(P_1,N)$     similar lexical/syntactic properties and semantically non-equivalent.
$(P_1,P_2)$   different lexical/syntactic properties and semantically equivalent.
$(P_2,N)$     different lexical/syntactic properties and semantically non-equivalent.

## D.1   Measuring Lexical/Syntactic Variations

Syntactic aspects of a sentence representing how words are put together to form constituents are best represented by its constituency parse trees. Accordingly, recent techniques, such as FastKASSIM [14] and CASSIM [9], utilize the constituency parse trees to estimate the syntactic similarity between a pair of sentences. On the other hand, lexical similarity between sentences can be measured by comparing the words used in each sentence and simple measures to compute lexical similarity include jaccard scores (i.e., word overlap) and edit-distances. We define the following metric to jointly measure the syntactic and lexical similarity (SLS) between two sentences $U$ and $V$:

$$SLS(U,V) = \text{LexicalSimilarity}(U,V) \times \text{SyntacticSimilarity}(U,V)$$

We use the normalized levenshtein similarity based on [96] as a measure of lexical similarity and FastKASSIM [14] with default parameters as a measure of syntactic similarity. We note that both of these metrics yield similarity scores in the range of and hence $SLS \in [0,1]$ closer to 1 indicates higher syntactic and lexical similarity.

For completeness, we extended this analysis and include a boxplot of the SLS in Figure 6. We find that the $(P_1, P_2)$ sentence pairs have consistently lower SLS scores as compared to $(P_1, N)$ sentence pairs: specifically, on average, we find that majority of the $(P_1, P_2)$ similarity scores lie between 0.2 - 0.4, while majority of the $(P_1, N)$ scores are greater than 0.75. In summary, this shows that (a) the sentences which we intended to be lexically/syntactically similar are indeed so, according to this measure; and (b) the sentences which we intended to be lexically/syntactically different are indeed so as well, according to this measure.

Below we summarize the lexical-similarity and syntactic-similarity:

(a) *Lexical-Similarity*: By definition, lexical refers to words/vocabulary and hence, lexical-similarity compares a pair of sentences at the word-level. In particular, a higher overlap of vocabulary and order of occurrence should lead to higher lexical similarity. We measure lexical-similarity using normalized Levenshtein similarity [96].
(b) *Syntactic-Similarity*: Syntax refers to the arrangement of words to form constituents (e.g., phrase/clause) and hence, syntactic-similarity compares a pair of sentences based on their constituency parse trees. Thus, syntactic similarity is higher for sentences whose constituency parse trees share a common structure. We use the FastKASSIM [14] score to measure syntactic-similarity.

Lexical similarity is more aligned with human perception as compared to syntactic-similarity but does not consider grammatical nuances while estimating similarity. On the other hand, constituency

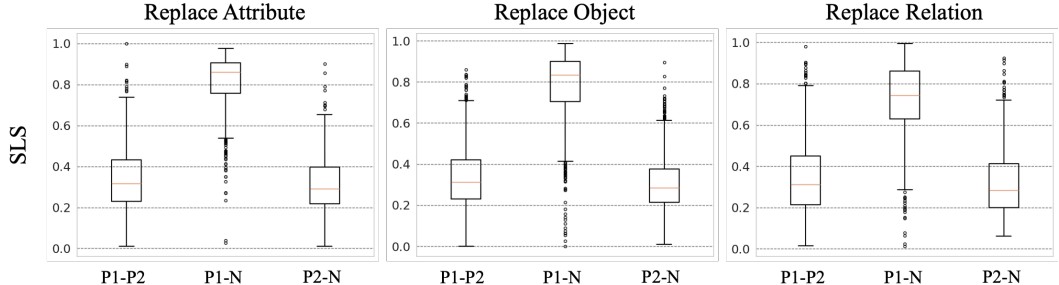

Figure 6: Box plot of syntactic and lexical similarity (SLS) for caption pairs in SUGARCREPE++. $P_1$ and $P_2$ refer to positive captions, while $N$ refers to the negative caption. $P_1$-$P_2$, $P_1$-$N$, and $P_2$-$N$ represent the distribution of SLS for the corresponding caption pairs, respectively. We find that the $P_1$-$P_2$ sentence pairs consistently have lower SLS scores compared to the $P_1$-$N$ sentence pairs.

parse trees are constructed using rules of grammar and seemingly minor changes to a sentence can introduce important structural changes in the parse-tree. For example, consider a simple example (not from our dataset):

Sentence-1  A[Article] boy[Noun] sits[Verb]
Sentence-2  The[Article] girl[Noun] runs[Verb]

The above two sentences are semantically different i.e., they convey very different meanings but the syntactic structure of these two sentences is identical. We provide examples of sentence-pairs from our dataset along with their syntactic-similarity, lexical-similarity scores and SLS scores in Table 9.

Table 9: Example caption pairs from SUGARCREPE++ with their corresponding syntactical-similarity, lexical-similarity and SLS scores.

| Sentence 1 | Sentence 2 | Syntactic Similarity | Lexical Similarity | SLS |
|---|---|---|---|---|
| A living room with white furniture and a small wooden table. | A living room without white furniture and a small wooden table. | 1.00 | 0.95 | 0.95 |
| Three adult zebras walk calmly along close together. | Three adult horses walk calmly along close together. | 1.00 | 0.91 | 0.91 |
| A street light in front of a colorful train on a bridge. | A colorful train is on a bridge with a street light in front of it. | 0.84 | 0.28 | 0.24 |
| A teddy bear is placed on a metallic sculpture. | The metallic sculpture is positioned below the teddy bear. | 1.00 | 0.24 | 0.24 |
| A desk with two computer monitors, two mice, a cup and a keyboard on it. | There are two computer monitors, two mice, a cup, and a keyboard on the desk. | 0.26 | 0.76 | 0.20 |
| A fire hydrant is decorated with an American flag design. | The American flag design is adorned on the fire hydrant. | 0.96 | 0.19 | 0.18 |
| An empty clean kitchen with cabinetry, stove and dishwasher. | An empty kitchen featuring cabinets, stove, and a dishwasher is clean. | 0.30 | 0.60 | 0.18 |
| A table topped with apples, oranges and bananas. | The table stands as a backdrop to a fruitful display, showing apples, oranges, and bananas arranged on top. | 0.11 | 0.40 | 0.04 |
| A white chair, books and shelves and a tv on in this room. | In this room, there is a white chair, shelves, books, and a TV on. | 0.12 | 0.27 | 0.03 |

# E  Additional Analysis of VLMs

## E.1  Specification of Evaluated VLMs

Table 10: Details of the VLMs evaluated on the SUGARCREPE++ dataset. Pretraining Data type: R, N and S refer to Real, Noisy and Synthetic data types, respectively. Pretraining Objectives – ITC: image-text contrastive; ITM: image-text matching; MLM: masked language modeling; MMM: masked multimodal modeling; MIM: masked image modeling; IC: image captioning; IS: image segmentation using KL divergence; ITA: image-text alignment; CCL: Cycle-consistency loss; finetuning objectives – ITR: image-text retrieval; NL: Negative loss for text; SG: scene graph loss; PT, FT refer to pretraining and finetuning, respectively. CLIP refers to CLIP-ViT-B-32 model.

| Model | #Total Parameters | Embedding Dimension | Pretraining Data size | Pretraining Data Type | Pretraining Objectives | Finetuned |
|---|---|---|---|---|---|---|
| CLIP 2021 | 151M | 512 | 400M | R | ITC | ✗ |
| RoBERTa-ViT-B-32 2022 | 212M | 512 | 2B | R | ITC | ✗ |
| ALIGN 2021 | 490M | 640 | 1.8B | R+N | ITC | ✗ |
| ALIP 2023 | 151M | 512 | 15M | R+S | ITC | ✗ |
| FLAVA 2022 | 358M | 768 | 70M | R | ITC, ITM, MLM MMM, MIM | ✗ |
| ALBEF 2021 | 210M | 256 | 14M | R+N | ITC, ITM, MLM | ✗ |
| BLIP 2022 | 225M | 512 | 129M | R+S | ITC, ITM, IC | ✗ |
| BLIP2 2023 | 1173M | 256 | 129M | R+S | ITC, ITM, IC | ✗ |
| ViLT 2021 | 111M | 768 | 10M | R | ITM, MLM | ✗ |
| AltCLIP 2023 | 864M | 768 | 42M | R | ITC | ✗ |
| SegCLIP 2023 | 151M | 512 | 400M+4M | R | ITC, MIM, IS | ✗ |
| XVLM-4M 2022 | 216M | 256 | 4M | R | ITC, ITM, MLM, ITA | ✗ |
| XVLM-16M 2022 | 216M | 256 | 16M | R | ITC, ITM, MLM, ITA | ✗ |
| ViLT-ITR-COCO 2021 | 111M | 768 | PT: 10M FT: 110K | R | ITM, MLM FT: ITR | ✓ |
| XVLM-16M-COCO 2022 | 216M | 256 | PT:16M FT: 110K | R | ITC, ITM, MLM, ITA FT: ITR | ✓ |
| XVLM-16M-Flickr 2022 | 216M | 256 | PT: 16M FT: 30K | R | TC, ITM, MLM, ITA FT: ITR | ✓ |
| NegCLIP 2023 | 151M | 512 | PT: 400M FT:110K | R | ITC FT: ITM | ✓ |
| CLIP-SVLC 2023 | 151M | 512 | PT:400M FT:400M | R | ITC FT: ITC, NL | ✓ |
| BLIP-SGVL 2023 | 696M | 768 | PT: 129M FT:4M | R | ITC, ITM, IC FT: ITC, SG | ✓ |
| CyCLIP 2022 | 102M | 1024 | PT: 102M | R | ITC, CCL | ✗ |

We comprehensively evaluate a wide array of VLMs on SUGARCREPE++ including (Table 10 provides details about different VLMs):

- Models trained with a contrastive learning objective such as CLIP [66], RoBERTa-ViT-B-32 [76], ALIGN [35] and ALIP [95]. ALIGN and ALIP utilize noisy and synthetic captions, respectively.
- Models trained by combining multiple objective functions, such as FLAVA [78]: pretrained by combining contrastive, Image-text matching (ITM), masked image modeling (MIM) and masked language modeling (MLM) objectives; ALBEF [43]: which combines ITM and MLM; BLIP [44] and BLIP-2 [45]: which combine contrastive, ITM and image captioning objectives.
- Models with a unified encoder for text and images, such as ViLT [38], and multi-lingual distilled models like AltCLIP [16].
- Models that align text with corresponding visual concepts in the image, such as Seg-CLIP [54], and XVLM [98] - with two variants, XVLM-4M and XVLM-16M.
- We also evaluate several models that have been finetuned on downstream tasks of image-text retrieval, such as ViLT-ITR-COCO [38] and XVLM-16M-ITR-COCO [98]. Specifically, ViLT, and XVLM-16M models were trained for the ITM task using the COCO dataset.

Additionally, XVLM-16M-ITR-Flickr [98] denotes XVLM-16M models trained for the ITM task using the Flickr dataset.

- Moreover, we evaluate recent methods proposed to improve the compositionality of VLMs, including NegCLIP [97], SVLC [21], CyCLIP [23], and BLIP-SGVL [26].

## E.2 Evaluation of Variants of CLIP.

Table 11: Details of different variants of CLIP that are evaluated on the SUGARCREPE++ dataset. Data, Model and Emb. refer to the pre-training dataset size and total number of parameters in the model (in Millions) and embedding dimension, respectively.

| Model | Pre-training Dataset | Pre-training Data size | # Params Model | Embed. Dimen. |
|---|---|---|---|---|
| RN50 [66] | WebImageText | 400M | 102M | 1024 |
| RN101 [66] | WebImageText | 400M | 120M | 512 |
| CLIP [66] | WebImageText | 400M | 151M | 512 |
| RN50×4 [66] | WebImageText | 400M | 178M | 640 |
| RN50×16 [66] | WebImageText | 400M | 291M | 768 |
| CLIP-ViT-L/14 [66] | WebImageText | 400M | 428M | 768 |
| RN50×64 [66] | WebImageText | 400M | 623M | 1024 |
| RoBERTa-ViT-B/32 [76] | LAION | 2B | 212M | 512 |
| ViT-H/14 [76] | LAION | 2B | 986M | 1024 |
| ViT-g/14 [76] | LAION | 2B | 1367M | 1024 |
| ViT-bigG/14 [76] | LAION | 2B | 2540M | 1280 |
| xlm-roberta-base-ViT-B/32 [76] | LAION | 5B | 366M | 512 |
| xlm-roberta-large-ViT-H/14 [76] | LAION | 5B | 1193M | 1024 |
| large:ViT-B/16 [22] | DataComp | 1B | 150M | 512 |
| xlarge:ViT-L/14 [22] | DataComp | 13B | 428M | 768 |

We evaluated variants of CLIP [66] that different in pre-training data size, model architecture and model size as listed below (see Table 11 for more details).

- CLIP [66] variants trained on the WebImageText dataset, which comprises 400 million image-text pairs. These models encompass CNN-based architectures, such as RN50, RN101, $RN50 \times 4$, $RN50 \times 16$, and $RN50 \times 64$, as well as transformer-based models like ViT-B/32 and ViT-L/14.
- CLIP-based models introduced by [76] pre-trained on extensive paired image-text datasets. Schuhmann et al. [76] provided diverse CLIP variants, namely RoBERTa-ViT-B/32, ViT-H/14, ViT-g/14, xlm-roberta-base-ViT-B/32, and xlm-roberta-large-ViT-H/14, trained on a large image-text dataset called "LAION-5B," which consists of 5 billion image-text pairs.
- Gadre et al. [22] released two CLIP variants, namely Large:ViT-B/16 and xlarge:ViT-L/14, trained on the DataComp dataset, comprising 13 billion image-text pairs.

Performance of various CLIP variants on SUGARCREPE++ dataset is provided in Table 12.

## E.3 Finetuning on VLMs on SUGARCREPE++

For the fine-tuning results, we divided the SUGARCREPE++ dataset into train and test sets. We used 40% of the randomly selected samples from each subset for training the models, and remaining 60% samples from each subset to obtain the performance of the models on each subset. We fine-tuned the models using objective function employed in training sentence transformers [70] to ensure a fair comparison between the fine-tuned and zero-shot evaluations of the models. During inference, we adhered to the approach described in our paper. Table 13 shows the performance of three different VLMs (CLIP-ViT-B/32, XLM-RoBERTa–ViT-B/32 and FLAVA) under the Frozen, full fine-tuning and LORA fine-tuning conditions. Experimental results show that even though there are improvements in performance on some subsets, fine-tuning the models can lead to degradation in

Table 12: Comparison of the performance of different variants of CLIP on SUGARCREPE++. Performance reported in terms of Accuracy (%). Overall best values are in bold, and group-level best values are underlined.

| Model | Swap Object | | Swap Attribute | | Replace Object | | Replace Attribute | | Replace Relation | |
|---|---|---|---|---|---|---|---|---|---|---|
| | ITT | TOT | ITT | TOT | ITT | TOT | ITT | TOT | ITT | TOT |
| RN50 [66] | 46.05 | 17.98 | 49.01 | 31.35 | 87.41 | 82.45 | 67.39 | 57.36 | 56.12 | 39.04 |
| RN101 [66] | 42.10 | 18.86 | 48.10 | 29.68 | 88.20 | 83.17 | 67.89 | 55.21 | 53.20 | 39.76 |
| CLIP [66] | 45.18 | 19.74 | 45.21 | 33.03 | 86.80 | 83.72 | 65.61 | 59.14 | 56.26 | 38.62 |
| RN50×4 [66] | 46.93 | 21.49 | 46.42 | 30.59 | 87.77 | 80.87 | 67.51 | 53.93 | 53.91 | 38.55 |
| RN50×16 [66] | 39.04 | 17.55 | 46.42 | 30.29 | 89.10 | 76.57 | 65.74 | 49.87 | 53.41 | 38.19 |
| CLIP-ViT-L/14 [66] | 43.86 | 17.11 | 45.36 | 28.92 | 90.68 | 80.69 | 67.39 | 55.96 | 54.05 | 39.26 |
| RN50×64 [66] | 44.74 | 16.67 | 45.36 | 31.51 | 90.79 | 73.31 | 64.47 | 48.61 | 54.27 | 38.12 |
| RoBERTa-ViT-B/32 [76] | 44.3 | 29.39 | 56.32 | 52.66 | 89.04 | 94.55 | 74.49 | 80.46 | 59.39 | 57.75 |
| ViT-H/14 [76] | 43.42 | 27.63 | 54.19 | 50.69 | 93.71 | 90.43 | 71.06 | 73.98 | 56.62 | 51.92 |
| ViT-g/14 [76] | 44.3 | 26.32 | 52.06 | 43.99 | 93.1 | 91.83 | 71.19 | 73.73 | 57.54 | 52.56 |
| ViT-bigG/14 [76] | 45.61 | 29.82 | 57.38 | 52.05 | 94.13 | 90.44 | 76.40 | 72.84 | 59.45 | 53.49 |
| xlm-RoBERTa-base-ViT-B/32 [76] | 42.55 | 30.26 | 55.25 | 55.56 | 89.41 | 95.34 | 72.97 | 80.96 | 55.48 | 57.82 |
| xlm-RoBERTa-large-ViT-H/14 [76] | 42.1 | 29.83 | 54.19 | 52.36 | 93.94 | 94.25 | 75.38 | 79.19 | 58.75 | 60.69 |
| large:ViT-B/16 [22] | 35.96 | 17.98 | 39.58 | 30.44 | 87.83 | 90.5 | 67.89 | 70.81 | 50.36 | 39.04 |
| xlarge:ViT-L/14 [22] | 42.55 | 26.75 | 46.58 | 39.12 | 91.59 | 91.89 | 72.59 | 71.45 | 55.83 | 49.79 |

Table 13: Image-to-text (ITT) results on SUGARCREPE++ under frozen, full-tuning and LoRA settings.

| | CLIP-ViT-B/32 | | | XLM-RoBERTa–ViT-B/32 | | | FLAVA | | |
|---|---|---|---|---|---|---|---|---|---|
| | Frozen | Fine-tuning | LoRA | Frozen | Fine-tuning | LoRA | Frozen | Fine-tuning | LoRA |
| Swap Object | 47.53 | 51.23 | 53.4 | 42.59 | 46.31 | 49.63 | 54.32 | 51.02 | 55.18 |
| Swap Attribute | 44.77 | 51.04 | 53.29 | 55.68 | 57.68 | 60.73 | 61.13 | 64.47 | 65.44 |
| Replace Object | 86.43 | 85.54 | 87.91 | 88.07 | 87.55 | 88.61 | 88.26 | 87.11 | 89.56 |
| Replace Attribute | 65.26 | 68.74 | 70.31 | 72.55 | 69.43 | 71.92 | 72.74 | 70.03 | 75.58 |
| Replace Relation | 57.65 | 60.49 | 61.15 | 58.1 | 59.38 | 60.82 | 59.91 | 63.53 | 62.21 |

performance on some subsets. This can be attributed to the catastrophic forgetting issue. To resolve this issue, we used the low-rank adapters (LoRA) [30], where we freeze the model's weights and train only the newly introduced LoRA parameters on our dataset. Here we report results for LoRA with rank = 2. It can be observed from the results that LoRA improves performance on our datasets across all subsets

Table 14: Few shot (4-shot and 8-shot) performance of ChatGPT-4o on SUGARCREPE++.

| | Swap Object | Swap Attribute | Replace Object | Replace Attribute |
|---|---|---|---|---|
| Zero-shot | 67.61 | 85.82 | 96.25 | 93.27 |
| 4-shot | 49.24 | 61.49 | 79.30 | 75.71 |
| 8-shot | 52.39 | 62.11 | 80.21 | 74.08 |

### E.4  Few-shot Learning on SUGARCREPE++

Table 14 provides the preliminary results on GPT-4o under a few-shot setting (4/8 shot). For these experiments, we provided demonstration samples (4-shot and 8-shot) from the same subset in addition to the query prompt – "Do any of these captions fail to correspond with the image? (1) <N> ; (2) <$P_1$>; (3) <$P_2$>; provide output as (1), (2) or (3)". We observed significant degradation in performance of GPT-4o in both 4-shot and 8-shot settings. We suspect this could be due to the choice of the reference samples or due to recency effects i.e., position of negative caption in the last reference sample influencing the decision of the VLM [101]. We also observed that GPT-4o is very sensitive to the type of examples given as reference. When samples from other subsets are provided as reference samples, further degradation in performance was observed. Future research can explore the options of tuning the prompt and to find the set of reference samples which could make few-shot learning effective on our dataset.

Although we conducted experiments with fine-tuning of VLMs (noting improvements in performance using LoRA) and few-shot learning, we emphasize the importance of the zero-shot evaluation procedure on SUGARCREPE++, which is followed through out the paper. The main objective of our dataset is to assess a fundamental property of VLMs: whether these models can correctly encode semantics regardless of the lexical/syntactic properties of the sentence. We believe this capability should be inherently learned by the models during the pre-training process. Fine-tuning on our dataset may enhance performance on our task but could also lead to catastrophic forgetting, impacting the model's performance on other downstream tasks.

### E.5  Comparison of Performance Between SUGARCREPE and SUGARCREPE++

Table 15, the extended version of Table 6, provides the comparison between SUGARCREPE ($P_1$ vs N), $P_2$ vs N and SUGARCREPE++ on the ITT task. Here $P_2$ vs N refers to the case where the model need to match the input image to the correct caption given $P_2$ (second positive caption) and N (negative caption) as options. As a direct comparison of the models' absolute performance on SUGARCREPE, $P_2$ vs N and SUGARCREPE++ is not possible, we assess their relative rankings using CLIP as a baseline. It can be observed that there are significant variations in the ordering of the models (comapred to CLIP performance) between the three cases. This shows that SUGARCREPE, and SUGARCREPE++ evaluate different aspects of the VLMs.

Table 15: Expanded version of Table 6 by including $P_2$vsN results. ↑ and ↓ show increases and decreases in performance with the corresponding CLIP performance as the baseline.

| Model | Swap Object | | | Swap Attribute | | | Replace Object | | | Replace Attribute | | | Replace Relation | | |
|---|---|---|---|---|---|---|---|---|---|---|---|---|---|---|---|
| | SC | $P_2$vsN | SC++ | SC | $P_2$vsN | SC++ | SC | $P_2$vsN | SC++ | SC | $P_2$vsN | SC++ | SC | $P_2$vsN | SC++ |
| CLIP-ViT-B/32 | 59.21 | 65.35 | 45.18 | 64.99 | 60.27 | 45.21 | 90.86 | 90.38 | 86.8 | 80.33 | 73.22 | 65.61 | 70.48 | 68.63 | 56.26 |
| ALBEF | 63.16↑ | 37.72↓ | 28.94↓ | 69.25↑ | 41.70↓ | 36.83↓ | 93.04↑ | 77.60↓ | 76.27↓ | 84.65↑ | 59.39↓ | 56.35↓ | 77.60↑ | 52.21↓ | 47.80↓ |
| XVLM | 64.91↑ | 42.10↓ | 31.14↓ | 73.97↑ | 46.27↓ | 36.52↓ | 95.22↑ | 80.87↓ | 79.42↓ | 87.69↑ | 62.56↓ | 59.39↓ | 77.45↑ | 51.78↓ | 46.23↓ |
| BLIP | 66.22↑ | 60.53↓ | 47.37↑ | 76.25↑ | 67.88↑ | 60.58↑ | 96.55↑ | 93.70↑ | 92.62↑ | 81.98↑ | 76.02↑ | 72.08↑ | 68.35↓ | 67.21↓ | 56.76↑ |
| NegCLIP | 75.44↑ | 64.47↓ | 55.25↑ | 76.87↑ | 63.93↑ | 57.99↑ | 93.88↑ | 91.16↑ | 89.53↑ | 87.18↑ | 72.84↓ | 69.41↑ | 74.47↑ | 58.32↓ | 52.27↓ |

## F  Additional Analysis of ULMs

Unimodal Language Models (ULMs) can be evaluated for their semantic and lexical sensitivity using the Text-only task (TOT) in SUGARCREPE++. We report the TOT results of ULMs in Table 16. We cover a comprehensive list of ULMs, varying in parameter counts, optimization objectives, training data sizes, and architectures. We notice that ULMs consistently fail to disassociate semantics from different lexical forms. This is indicated by the large deviation (ranging from 18% to 35%) in TOT accuracy across different subsets. Generally, all models perform over 90% accuracy in identifying the positive captions when the negative caption changes the type of an 'object' in the positive caption (altering the semantics and lowering lexical overlap). On the contrary, if multiple 'objects' are swapped within the same caption (altering the semantics but preserving the lexical overlap), the performance, on average across all ULMs, decreases by ≈50%. This can be attributed to the lack of compositional understanding required to associate semantics from different lexical forms.

Grouping the models based on model sizes reveals that sensitivity to lexical alteration does not necessarily improve with scale. For example, a small-sized model like BAAI General Embedding (BGE) [93] is only 5% behind very large LLMs like E5-mistral [89, 88] with 7 billion parameters in the 'Replace Relation' subset. We experimented with recently proposed text encoder models like INSTRUCTOR [81] that aim to improve the generalization of sentence embeddings across various embedding-based tasks. These models allow to add special prefix instructions before encoding text to condition for a task. We compared the INSTRUCTOR model with a default prefix and a prefix with instructions to encode the semantics (listed as custom-ins in Table 16). We notice minimal performance gains with custom instruction, demonstrating that further development is required to separate semantics from lexical composition.

Table 16: Comprehensive results of ULMs on TOT of SUGARCREPE++. We report the TOT accuracy (%). and group the results row-wise based on the model size as reflected by the parameter count. We include the number of parameters in text encoders relative to BERT-base, i.e., 109.5 Million parameters. Overall, best values are in bold, and group-level best values are underlined. We report the average and standard deviation across different subsets as an additional column.

| Model | #Params (BERT Scale) | Swap Object | Swap Attribute | Replace Object | Replace Attribute | Replace Relation | Average |
|---|---|---|---|---|---|---|---|
| All-MiniLM-L6-v2 [91] | 0.21 | 14.29 | 22.52 | 93.95 | 64.97 | 63.94 | $51.93_{33.02}$ |
| BGE-small-en-v1.5 [93] | 0.3 | 15.51 | 24.02 | 94.19 | 75.00 | 75.53 | $56.85_{34.85}$ |
| All-MiniLM-L12-v2 [91] | 0.3 | 18.78 | 25.38 | 95.22 | 73.86 | 70.41 | $56.73_{33.11}$ |
| GTE-small [47] | 0.3 | 13.88 | 22.07 | 94.98 | 71.95 | 69.06 | $54.39_{34.84}$ |
| Angle-BERT-base-uncased-nli-en-v1 [46] | 1 | 25.71 | 33.63 | 92.07 | 78.43 | 75.32 | $61.03_{29.45}$ |
| BGE-base-en-v1.5 [93] | 1 | 17.14 | 25.23 | 93.83 | 78.55 | 76.10 | $58.17_{34.56}$ |
| Sentence-T5-base [62] | 1.01 | 28.98 | 31.98 | 92.37 | 75.00 | 75.32 | $60.73_{28.51}$ |
| GTE-base [47] | 1 | 17.14 | 22.82 | 94.31 | 75.00 | 71.98 | $56.25_{34.26}$ |
| Instructor-large [81] | 3.07 | 26.53 | 28.38 | 96.00 | 72.34 | 73.83 | $59.42_{30.65}$ |
| Instructor-large(custom-ins)[81] | 3.07 | 22.86 | 27.63 | 96.13 | 77.41 | 77.81 | $60.37_{32.99}$ |
| UAE-Large-v1 [46] | 3.06 | 40.41 | 41.44 | **96.85** | 76.14 | 75.82 | $66.13_{24.54}$ |
| GTE-large [47] | 3.06 | 26.53 | 27.93 | 96.31 | 76.78 | 72.83 | $60.07_{31.28}$ |
| All-RoBERTa-large-v1 [70] | 3.25 | 42.04 | 45.20 | 94.61 | 74.75 | 74.96 | $66.31_{22.26}$ |
| Stsb-RoBERTa-large [70] | 3.25 | 25.31 | 31.98 | 94.19 | **89.21** | 75.18 | $63.17_{32.37}$ |
| Sentence-T5-xl [62] | 11.34 | **47.35** | **49.25** | 90.98 | 75.38 | 75.32 | $67.66_{18.8}$ |
| Angle-Llama-7b-nli-v2 [46] | 62.28 | 37.96 | 45.80 | 95.22 | 84.39 | **81.44** | $\mathbf{68.96}_{25.4}$ |
| E5-Mistral-7b-instruct [89, 88] | 64.95 | 33.47 | 37.84 | 96.67 | 87.06 | 80.51 | $67.11_{29.33}$ |

Table 17: Prompt-sensitivity analysis of BLIP (a generative VLM) using SUGARCREPE++. Results show that simple paraphrases of the input prompts effect model performance significantly.

| | Input Prompts | S. Obj | S. Att | R. Obj | R. Att | R. Rel |
|---|---|---|---|---|---|---|
| Prompt-1 Paraphrases | Does the following caption match the image: <Caption>? Provide 'yes' or 'no' as response | 6.1 | 18.12 | 48.09 | 34.64 | 16.06 |
| | Does the caption match the image: <Caption>? Provide 'yes' or 'no' as response | 7.34 | 20.87 | 45.46 | 37.56 | 18.55 |
| | Do the image and the following caption match: <Caption>? Provide 'yes' or 'no' as response | 8.21 | 12.46 | 40.86 | 29.06 | 11.95 |
| Prompt-2 Paraphrases | Does the caption correctly describe the image: <Caption>? Provide 'yes' or 'no' as response | 6.53 | 24.47 | 51.51 | 41.16 | 23.25 |
| | Is the caption an accurate description of the image: <Caption>? Provide 'yes' or 'no' as response. | 7.34 | 23.27 | 53.39 | 42.89 | 25.1 |

# G   Additional Results on Generative VLMs

## G.1   Prompt-based Evaluation of BLIP and BakLLaVA

We performed two different experiments to evaluate prompt sensitivity of generative VLMs using SUGARCREPE++.

1. **Paraphrasing of prompts:** We prompt BLIP (in VQA format) with paraphrases of the same prompt (see Table 17 for prompt details and results). It can be observed from table 17 that for different paraphrases of the same prompt, there are huge differences in model performance, for both Prompt 1 and Prompt 2. Moreover, there is no single variant of the prompt that achieved best performance across all subsets of SUGARCREPE++.

2. **Reordering options in the prompts:** As BLIP cannot handle long prompts, we analyzed BakLLaVA [53], a generative VLM based on Mistral 7B augmented with LLaVA architecture. We prompt BakLLaVA using the prompts shown in Figure 7, and report results on SUGARCREPE++ in Table 18. We observed that by simply changing the ordering of the captions, the performance of the BakLLaVA model changes drastically. This sensitivity of model performance to simple paraphrases of the prompt further highlights the importance of using our SUGARCREPE++ dataset to evaluate VLMs ability to understand semantic and lexical alterations.

| | | |
|---|---|---|
| **NPP:** | "*USER*: <image>\n Which one of the three captions does not match the image? (1) Negative; (2) Positive-1; (3)Positive-2; provide output as either (1), (2), or (3). Do not provide any explanation" "\n*ASSISTANT*:" – Answer will be (1) | |
| **PNP:** | "*USER*: <image>\n Which one of the three captions does not match the image? (1) Positive-1; (2) Negative; (3)Positive-2; provide output as either (1), (2), or (3). Do not provide any explanation" "\n*ASSISTANT*:" – Answer will be (2) | |
| **PPN:** | "*USER*: <image>\n Which one of the three captions does not match the image? (1) Positive-1; (2)Positive-2; (3) Negative; provide output as either (1), (2), or (3). Do not provide any explanation" "\n*ASSISTANT*:" – Answer will be (3) | |

Figure 7: Prompt variations considered by permuting order of options.

Table 18: Prompt sensitivity analysis of BakLLaVA on SUGARCREPE++. Here we experiment with reordering the position of Positive caption 1, Positive caption 2 and Negative caption in the prompt:

| BakLLaVA | Swap Obj | Swap Att | Replace Obj | Replace Att | Replace Rel |
|---|---|---|---|---|---|
| Negative, Positive1, Positive 2 | 39.94 | 49.71 | 61.99 | 36.71 | 41.26 |
| Positive 1, Negative, Positive 2 | 33.52 | 47.18 | 59.41 | 41.5 | 37.45 |
| Positive 1, Positive 2, Negative | 68.08 | 78.12 | 83.23 | 73.84 | 75.96 |

## G.2 Generative VLM Performance Using VGPTScore and VQAScore

We consider five additional generative VLMs and evaluate two recent methods — VGPTScore [50] and VQAScore [51, 42] — on image-text task of the SUGARCREPE++ dataset. For reference, we also include the corresponding results on SUGARCREPE (See Table 19). Both VGPTScore and VQAScore are methods designed to make the most effective use of generative VLMs for image-text matching tasks. We further note that the additionally considered generative VLMs range between 3B-11B in parameter counts and are larger than most VLMs previously considered in the paper — from Tables 10 and 11, we can see that the largest model we considered is a 2.5B parameter CLIP model (ViT-bigG/14). Despite using significantly larger models, we find that VGPTScore still performs comparably with most other models already included in the paper (e.g., Table 2). However, using the same generative VLMs, VQAScore achieves significantly improved performance. Nevertheless, we note that this performance is comparable with that of GPT-4o (Table 8) and in general, a significant gap still exists between the current performance and human-level performance.

Table 19: Generative VLM performance on ITT task of SUGARCREPE++ using VGPTScore [50], VGPTScore Blind [50], and VQAScore [51, 42]. We consider five generative VLMs and report their performance on different subsets of SUGARCREPE and SUGARCREPE++. The rows are grouped by the method used for identifying the correct caption and the VLM model. The columns represent the different datasets and their subsets. We report the accuracy of the image-text task for SUGARCREPE++, with average performance presented in an additional column. We obtain the performance of a blind version of VGPTScore using gaussian images.

| | Model | SUGARCREPE (ITT %) | | | | | | SUGARCREPE++ (ITT %) | | | | | |
|---|---|---|---|---|---|---|---|---|---|---|---|---|---|
| | | Swap Object | Replace Attribute | Replace Object | Replace Relation | Swap Attribute | Avg | Swap Object | Replace Attribute | Replace Object | Replace Relation | Swap Attribute | Avg |
| **VGPTScore** | Clip-FlanT5-xl | 77.14 | 79.44 | 80.33 | 74.40 | 82.43 | 78.75 | 60.41 | 61.55 | 65.31 | 54.91 | 71.02 | 62.64 |
| | Clip-FlanT5-xxl | 77.55 | 80.33 | 78.75 | 77.03 | 87.84 | 80.30 | 58.78 | 57.23 | 57.57 | 54.27 | 75.53 | 60.67 |
| | InstructBLIP | 76.73 | 83.50 | 87.11 | 89.19 | 86.49 | 84.60 | 54.69 | 51.78 | 63.92 | 53.13 | 58.26 | 56.36 |
| | Llava-v1.5-7b | 76.33 | 70.69 | 71.31 | 76.53 | 84.98 | 75.97 | 54.29 | 48.60 | 51.82 | 52.06 | 70.12 | 55.38 |
| | ShareGPT4v-7b | 79.59 | 73.98 | 74.39 | 80.51 | 88.89 | 79.47 | 59.59 | 51.40 | 53.69 | 56.97 | 75.08 | 59.35 |
| **VGPTScore Blind** | Clip-FlanT5-xl | 63.67 | 56.47 | 47.94 | 64.58 | 65.62 | 59.66 | 47.76 | 39.59 | 31.54 | 42.03 | 51.95 | 42.57 |
| | Clip-FlanT5-xxl | 61.63 | 57.99 | 48.24 | 64.86 | 66.82 | 59.91 | 42.04 | 39.85 | 30.08 | 42.18 | 50.90 | 41.01 |
| | InstructBLIP | 60.82 | 64.59 | 59.56 | 73.90 | 73.27 | 66.43 | 43.67 | 43.15 | 39.95 | 47.94 | 52.55 | 45.45 |
| | Llava-v1.5-7b | 57.14 | 51.14 | 42.13 | 65.01 | 65.92 | 56.27 | 42.04 | 35.53 | 28.45 | 45.23 | 52.70 | 40.79 |
| | ShareGPT4v-7b | 56.73 | 52.03 | 44.19 | 66.71 | 68.17 | 57.57 | 40.82 | 36.42 | 30.99 | 46.30 | 54.20 | 41.75 |
| **VQAScore** | Clip-FlanT5-xl | 80.82 | 89.85 | 96.85 | 83.29 | 90.09 | 88.18 | 72.24 | 86.68 | 95.34 | 77.67 | 85.74 | 83.53 |
| | Clip-FlanT5-xxl | 83.27 | 94.67 | 97.64 | 88.98 | 94.74 | 91.86 | 74.69 | 88.96 | 95.94 | 82.15 | 88.29 | 86.01 |
| | InstructBLIP | 78.37 | 91.62 | 96.73 | 84.57 | 84.68 | 87.19 | 60.00 | 83.38 | 93.77 | 76.32 | 74.02 | 77.50 |
| | Llava-v1.5-7b | 74.29 | 87.56 | 95.10 | 80.58 | 80.33 | 83.57 | 57.14 | 78.43 | 91.53 | 70.41 | 68.17 | 73.14 |
| | ShareGPT4v-7b | 80.82 | 93.15 | 97.88 | 86.98 | 89.64 | 89.69 | 66.94 | 83.25 | 94.37 | 75.11 | 74.62 | 78.86 |

# H Qualitative Results

We perform qualitative analysis to inspect examples from SUGARCREPE++ dataset, where majority of models fail to dissociate semantics from lexical composition. In this section, we highlight the results of our qualitative analysis.

## H.1 Both Image-Text Task (ITT) and Text-Only Task (TOT) Fail

In this subsection, we provide examples where majority of vision-language models failed in both the image-text task and the text-only task. These examples demonstrate cases where both the text encoder and vision encoder of VLMs were ineffective in encoding semantics. Figure 8 shows such examples from each subset of SUGARCREPE++ dataset.

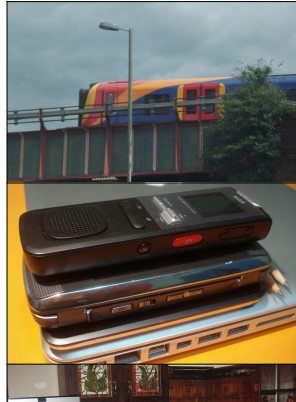

**Swap Object**

**P1:** A street light in front of a colorful train on a bridge.

**P2:** A colorful train is on a bridge with a street light in front of it.

**N:** A colorful train in front of a street light on a bridge.

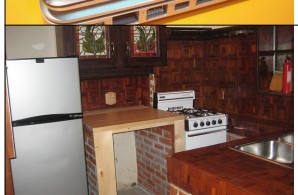

**Swap Attribute**

**P1:** There are two electronic devices on top of the laptop.

**P2:** The laptop is positioned below the two electronic devices.

**N:** There are electronic devices on top of the two laptops.

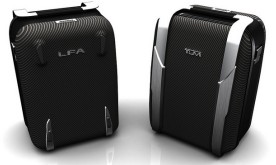

**Replace Object**

**P1:** Kitchen view with brick framework around the sink and by the oven

**P2:** The sink and oven are surrounded by a brick framework in the kitchen view.

**N:** Kitchen view with brick framework around the stove and by the oven.

**Replace Attribute**

**P1:** Two pieces of hard luggage are seen here.

**P2:** A couple of pieces of hard luggage are visible in this location.

**N:** Two pieces of soft luggage are seen here.

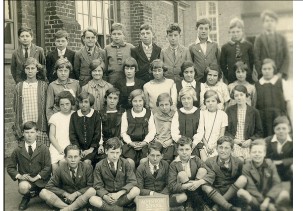

**Replace Relation**

**P1:** The black and white photograph of a classroom of schoolchildren is a bit out of focus on the right side of the picture.

**P2:** The photograph of a classroom of schoolchildren is slightly blurred on the right side of the image.

**N:** The black and white photograph of a classroom of schoolchildren is a bit out of focus on the left side of the picture.

Figure 8: Examples from each subset of SUGARCREPE++ dataset where majority of VLMs failed both the image-text task (ITT) and the text-only task (TOT).

## H.2   Image-Text Task (ITT) Pass and Text-Only Task (TOT) Fail

In this subsection, we provide examples where the majority of vision-language models (VLMs) failed in text-only task (TOT) but were successful in image-text task (ITT), indicating that the vision encoders were able to represent semantics more effectively than text encoders for these examples. Figure 9 shows such examples from each subset of SUGARCREPE++ dataset.

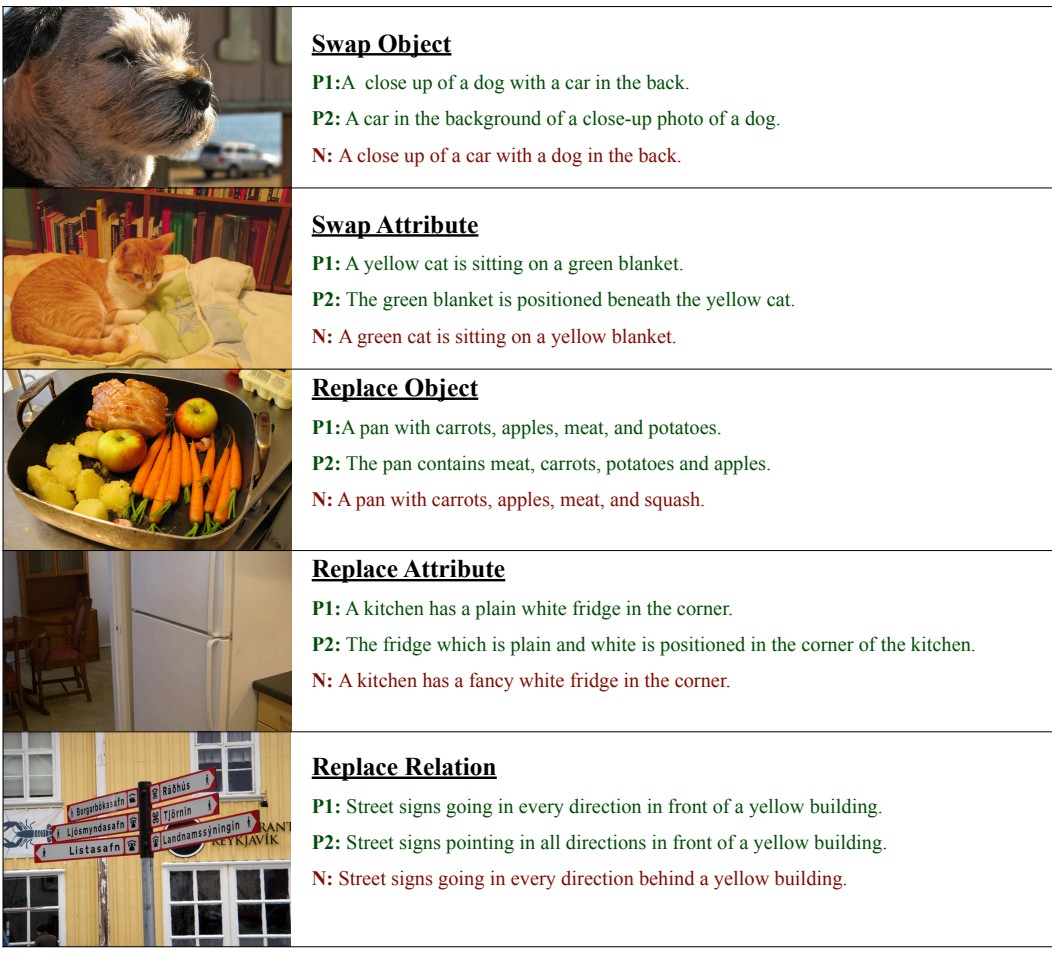

Figure 9: Examples from each subset of SUGARCREPE++ dataset where majority of VLMs passed the image-text task (ITT) and failed the text-only task (TOT).

### H.3 Text-Only Task (TOT) Pass and Image-Text Task (ITT) Fail

In this subsection, we provide examples where majority of vision-language models (VLMs) failed in the image-text task (ITT) but succeeded in the text-only task (TOT). These examples demonstrate cases where the text encoders of VLMs were more effective at encoding semantics than their vision encoders. Figure 10 presents examples from each subset of SUGARCREPE++ dataset.

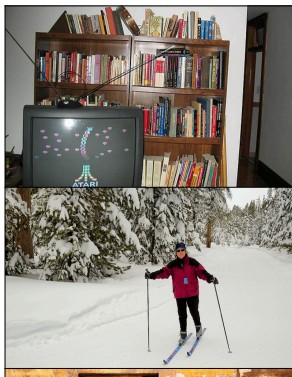

**Swap Object**

**P1:** An old television set is displaying an old computer game in front of two bookshelves.

**P2:** An old television set is positioned in front of two bookshelves and displaying an old computer game.

**N:** Two old bookshelves are displaying an old computer game in front of a television set.

**Swap Attribute**

**P1:** A person riding skis down a snow covered slope.

**P2:** A person is descending a snow-covered slope while riding skis.

**N:** A person covered in snow skis down a slope.

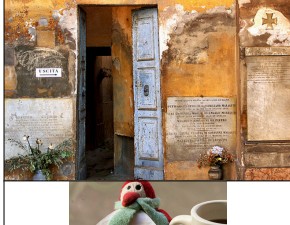

**Replace Object**

**P1:** A plaster external wall with multiple old paper images attached.

**P2:** An external wall made of plaster, adorned with several old paper images.

**N:** A wooden door with multiple old paper images attached.

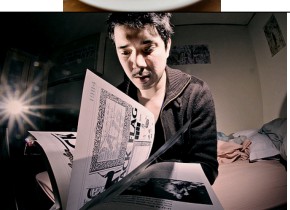

**Replace Attribute**

**P1:** A donut with a cup of coffee and an ornate napkin holder

**P2:** A cup of coffee along with an ornate napkin holder, and a donut.

**N:** A donut with a cup of coffee and a minimalist napkin holder.

**Replace Relation**

**P1:** A person sitting on a bed looking at a book.

**P2:** A person looking at a book while sitting on a bed.

**N:** A person leaning against a bed looking at a book.

Figure 10: Examples from each subset of SUGARCREPE++ dataset where majority of VLMs passed the text-only task (TOT) and failed the image-text task (ITT).

## I Additional Statistics for SUGARCREPE++

### I.1 Vera and Grammar Scores of SUGARCREPE++

Hsieh et al. [29] proposed an experiment based on Vera and Grammar scores to determine if language biases (e.g., fluency/grammar) can be exploited to "solve" the image-text matching task. We repeated this experiment on SUGARCREPE++ and visualized the score gaps (e.g., $0.5 \times (\text{Vera}(P1) + \text{Vera}(P2)) - \text{Vera}(N)$ ) in Figure 11. For an ideal dataset that cannot be solved by exploiting language bias, we should observe that both Grammar/Vera score gaps are centered around zero. Accordingly, we confirm that both SUGARCREPE++ as well as SUGARCREPE cannot be solved by exploiting language biases. We also provide numerical scores in Table 20.

Table 20: Comparison between SUGARCREPE++ and previous dataset in terms of Vera, Grammar, and VLM performance (averaged across 15 variants of CLIP)

| Dataset | Category | Random | VERA | Grammar | VLMs |
|---|---|---|---|---|---|
| ARO | VG-Relation | 50 | 61.71 | 59.55 | 50.53 |
| | VG-Attribution | 50 | 82.59 | 58.38 | 61.03 |
| | COCO-Order | 50 | 59.81 | 74.33 | 29.59 |
| | Flickr30K-Order | 50 | 63.52 | 76.26 | 35.93 |
| VL-CheckList | Object | 50 | 82.48 | 57.95 | 86.48 |
| | Attribute | 50 | 73.99 | 52.35 | 68.53 |
| | Relation | 50 | 85.72 | 68.5 | 70.7 |
| SUGARCREPE | Replace | 50 | 49.43 | 50.00 | 82.33 |
| | Swap | 50 | 49.30 | 50.00 | 64.82 |
| SUGARCREPE++ (ours) | Replace | 33.33 | 36.50 | 33.30 | 76.06 |
| | Swap | 33.33 | 40.00 | 34.60 | 46.33 |

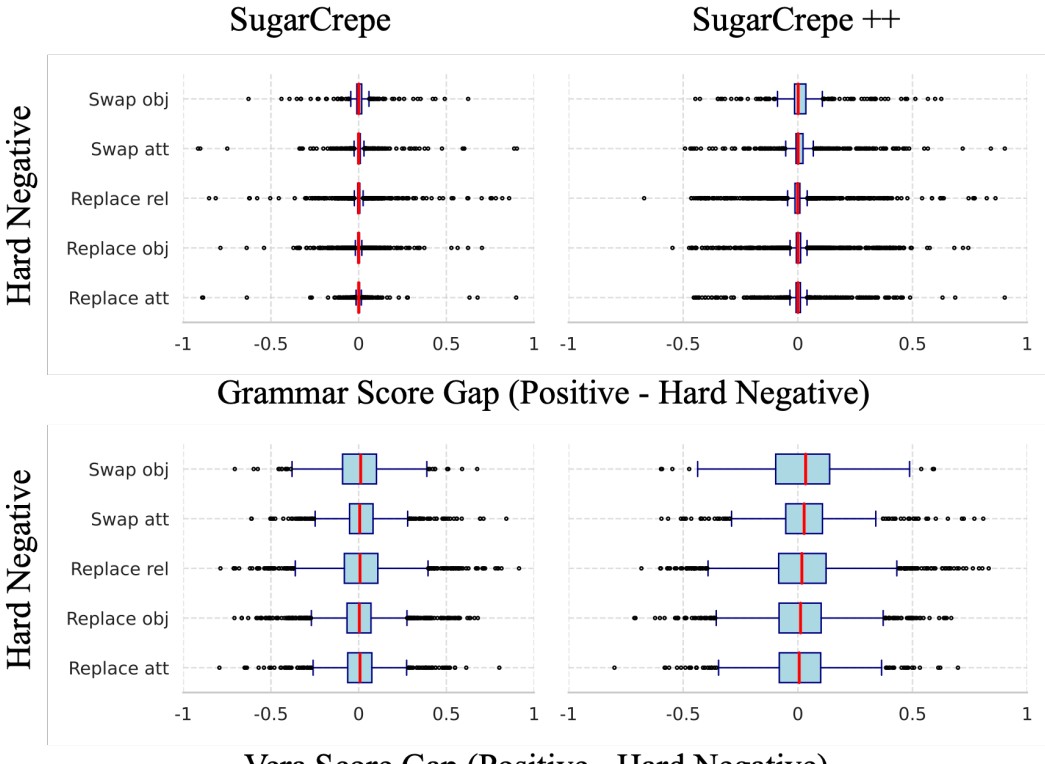

Figure 11: Vera and Grammar score gap plots for SUGARCREPE and SUGARCREPE++. The SUGARCREPE++ score gaps is computed as $0.5 \times (\text{Vera}(P1) + \text{Vera}(P2)) - \text{Vera}(N)$.

## I.2 MS-COCO Categories in SUGARCREPE++

We use images from MS-COCO [49, 10], which defines 91 "stuff" categories (amorphous, uncountable elements like "sky," "water") and 80 "object" categories (countable, distinct items like "car," "dog"), which are fully covered in SUGARCREPE++. We provide the distribution of these categories and their super-categories and subcategories in Figure 12 . Additionally, we identify key topics in the caption text using the BERTopic [25] package, and Figure 13 illustrates the top four topics across different subsets of SUGARCREPE++.

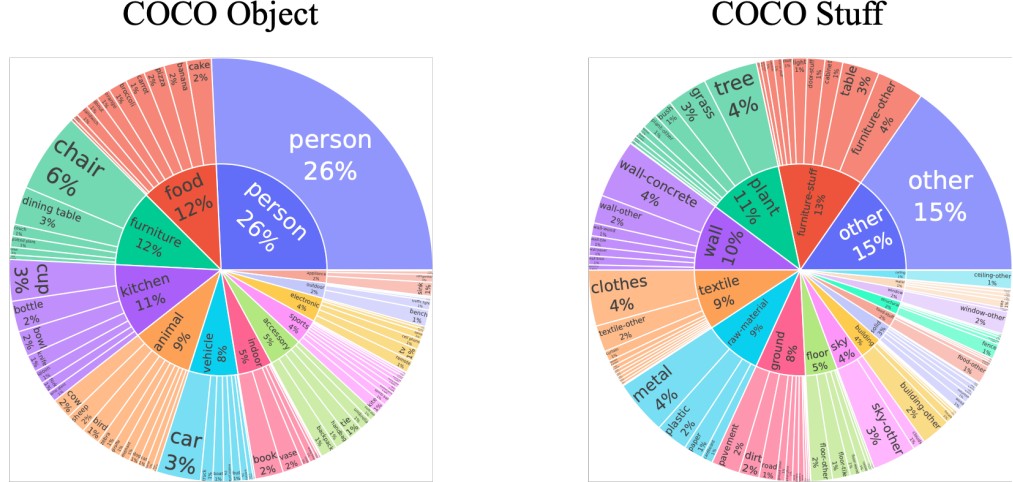

Distribution of COCO categories in SugarCrepe++

Figure 12: Distribution of MS-COCO [49, 10] super-categories and sub-categories in SUGAR-CREPE++.

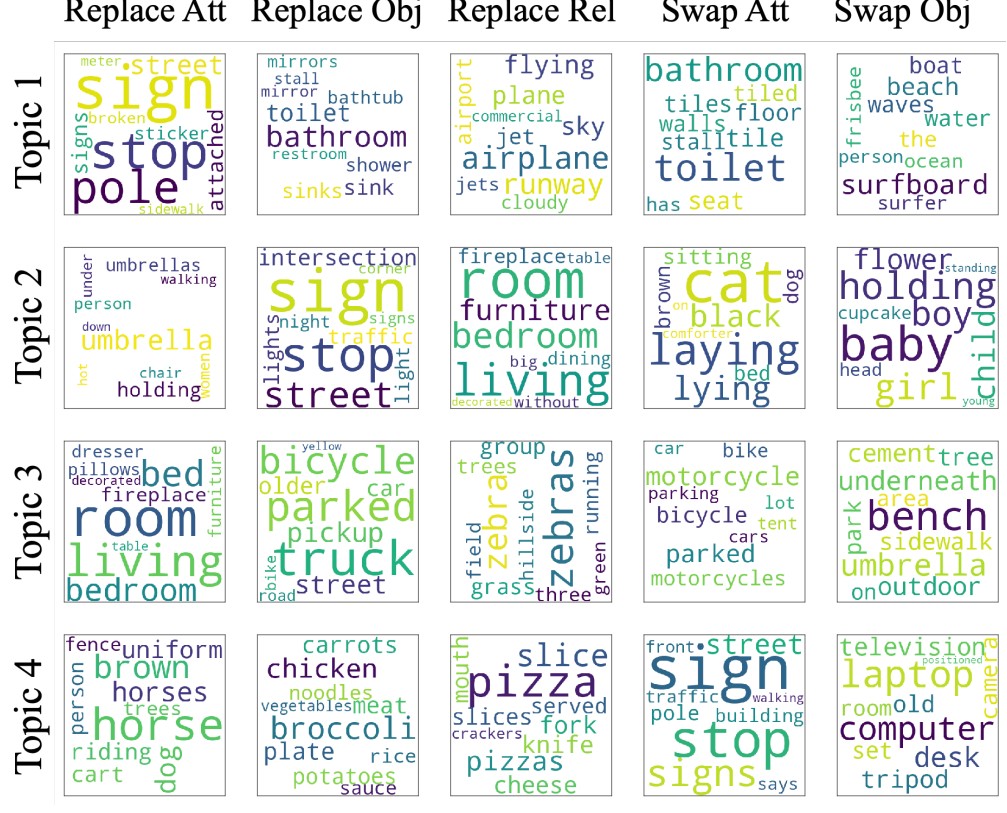

Figure 13: Topics in SUGARCREPE++ caption text, identified using BERTopic and illustrated using word clouds. We identify and illustrate the common topics present in the caption text of SUGARCREPE++ using word clouds. The different topics are represented vertically for all subsets of SUGARCREPE++. We show the top four topics among the multiple topics identified by the BERTopic package [25] across subsets in SUGARCREPE++.

## J  Human Evaluation Instruction

Figure 14 provides the instructions given to the human evaluators along with the screen shot of the interface used for human evaluation.

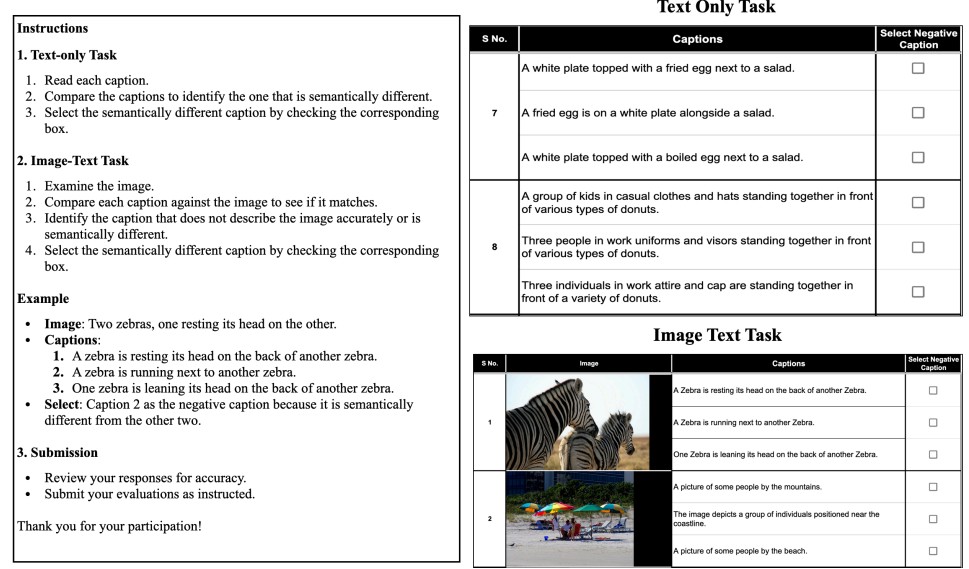

Figure 14: Human evaluation instructions and screenshot of the interface.

## K  Implementation Details

### K.1  Hardware Information

We performed all the experiments in this paper using a single 40G NVIDIA A100 GPU available in the Compute Canada Cluster.

### K.2  Dataset Sources

We obtain all existing datasets from their original sources released by the authors. We refer readers to these sources for questions regarding obtaining consent, dataset licenses and collection procedure.

- COCO [49]: We obtain COCO images from its official project website [5]. We use the images from the validation set [6]
- SUGARCREPE [29]: We obtain SUGARCREPE captions and hard negatives from its official website [7].

### K.3  Model Sources

**Evaluation of VLMs.**  Source and links of the VLMs detailed in Table 8 (in the Appendix of our paper) is provided below.

---

[5] https://cocodataset.org/
[6] http://images.cocodataset.org/zips/val2017.zip
[7] https://github.com/RAIVNLab/sugar-crepe

- CLIP [66]: 'ViT-B/32' variant of CLIP available at HuggingFace Link
- RoBERTa-ViT-B-32 [76]: RoBERTa-ViT-B-32 trained on LAION dataset available at HuggingFace Link
- ALIGN [35]: Model available at HuggingFace Link
- ALIP [95]: Model available at Google Drive Link
- FLAVA [78]: Model available at HuggingFace Link
- ALBEF [43]: ALBEF base model available in LAVIS
- BLIP [44]: BLIP base model available in LAVIS
- BLIP2 [45]: BLIP2 pretrained model available in LAVIS
- ViLT [38]: Pre-trained ViLT model available at HuggingFace Link
- SegCLIP [54]: Model available at Google Drive Link
- XVLM-4M [98]: XVLM base model trained using 4 Million samples available at Google Drive Link
- XVLM-16M [98]: XVLM base model trained using 16 Million samples available at Google Drive Link
- ViLT-ITR-COCO [38]: ViLT model finetuned for image-text retrieval task using MSCOCO dataset. This model is available at HuggingFace Link
- XVLM-16M-COCO [98]: XVLM 16 Million model fine-tuned for image-text-retrieval using MSCOCO dataset. This model is available at Google Drive Link
- XVLM-16M-Flickr [98]: XVLM 16 Million model fine-tuned for image-text-retrieval using Flickr dataset. This model is available at Google Drive Link
- NegCLIP [97]: NegCLIP is trained on top of CLIP and the model link is available in Github Link
- CLIP-SVLC [21]: Model is available at Google Drive Link
- BLIP-SGVL [26]: Model is available at Google Drive Link
- CyCLIP [23]: Model is available at Google Drive Link

**Variants of CLIP:** All the CLIP models' weights for the different variants of CLIP reported in Table 10 in the appendix are obtained from OpenCLIP [8] framework [32].

**Evaluated ULMs:** Source and links of the ULMs detailed in Table 11 (in the Appendix of out paper) is provided below.

- All-MiniLM-L6-v2 [91]: HuggingFace Link
- BGE-small-en-v1.5 [93]: HuggingFace Link
- All-MiniLM-L12-v2 [91]: HuggingFace Link
- GTE-small [47]: HuggingFace Link
- Angle-BERT-base-uncased-nli-en-v1 [46]: HuggingFace Link
- BGE-base-en-v1.5 [93]: HuggingFace Link
- Sentence-T5-base [62]: HuggingFace Link
- GTE-base [47]: HuggingFace Link
- Instructor-large [81]: HuggingFace Link
- Instructor-large (custom-ins)[81]: HuggingFace Link, we use *'Represent the sentence for spatial semantics'* as the custom instruction for Instructor-large (custom-ins) model.
- UAE-Large-V1 [46]: HuggingFace Link
- GTE-large [47]: HuggingFace Link
- All-RoBERTa-large-v1 [70]: HuggingFace Link

---

[8] https://github.com/mlfoundations/open_clip

- Stsb-RoBERTa-large [70]: HuggingFace Link
- Sentence-T5-xl [62]: HuggingFace Link
- Angle-Llama-7b-nli-v2 [46]: HuggingFace Link

## K.4    Reproducibility

We release SUGARCREPE++ dataset and the code to evaluate models on Github [9]. The datasheet for SUGARCREPE++ is provided in the Supplementary material and in the Appendix L. The HuggingFace dataset **croissant metadata** is available here.

## K.5    Author Statement

In case of violation of rights, the authors will bear all responsibility. We publicly release SUGAR-CREPE++ dataset under the **CC-BY-4.0** license.

## K.6    License, Hosting and Maintenance Plan

We release the dataset publicly under the **CC-BY-4.0** license on Github. The authors of this paper are committed to support and maintain the dataset via our GitHub repository.

---

[9] https://github.com/Sri-Harsha/scpp

# L Datasheet

## L.1 Motivation

Q1 **For what purpose was the dataset created?** Was there a specific task in mind? Was there a specific gap that needed to be filled? Please provide a description.

- The SUGARCREPE++ dataset was created to evaluate the sensitivity of vision language models (VLMs) and unimodal language models (ULMs) to semantic and lexical alterations. The SUGARCREPE dataset consists of (only) one positive and one hard negative caption for each image. Relative to the negative caption, a single positive caption can either have low or high lexical overlap. The original SUGARCREPE only captures the high overlap case. To evaluate the sensitivity of encoded semantics to lexical alteration, we require an additional positive caption with a different lexical composition. SUGARCREPE++ fills this gap by adding an additional positive caption enabling a more thorough assessment of models' abilities to handle semantic content and lexical variation.

Q2 **Who created the dataset (e.g., which team, research group) and on behalf of which entity (e.g., company, institution, organization)?**

- The SUGARCREPE++ dataset is created by the authors of this paper (affiliated to Faculty of Computer Science, Dalhousie University) to advance our understanding of language models through a new evaluation dataset/task.

Q3 **Who funded the creation of the dataset?** If there is an associated grant, please provide the name of the grantor and the grant name and number.

- We acknowledge the support provided by the Faculty of Computer Science, Dalhousie University. Resources used in preparing this research were provided, in part, by the support of the Natural Sciences and Engineering Research Council of Canada (NSERC), the Province of Ontario, the Government of Canada through Canadian Institute for Advanced Research (CIFAR), ACENET (ace-net.ca), the Digital Research Alliance of Canada (alliancecan.ca) and companies sponsoring the Vector Institute www.vectorinstitute.ai/#partners.

Q4 **Any other comments?**

- No.

## L.2 Composition

Q5 **What do the instances that comprise the dataset represent (e.g., documents, photos, people, countries)?** *Are there multiple types of instances (e.g., movies, users, and ratings; people and interactions between them; nodes and edges)? Please provide a description.*

- The instances from SUGARCREPE++ dataset represent images from MS-COCO [49] and their associated text captions, negative captions from SUGARCREPE and newly introduced positive captions.

Q6 **How many instances are there in total (of each type, if appropriate)?**

- In total, SUGARCREPE++ dataset consists of 4757 instances. The detailed statistics of the subcategories are provided in https://github.com/Sri-Harsha/scpp.

Q7 **Does the dataset contain all possible instances or is it a sample (not necessarily random) of instances from a larger set?** *If the dataset is a sample, then what is the larger set? Is the sample representative of the larger set (e.g., geographic coverage)? If so, please describe how this representativeness was validated/verified. If it is not representative of the larger set, please describe why not (e.g., to cover a more diverse range of instances, because instances were withheld or unavailable).*

- We included all possible instances from the SUGARCREPE dataset, except those which are not suitable for our tasks.

Q8 **What data does each instance consist of?** *"Raw" data (e.g., unprocessed text or images) or features? In either case, please provide a description.*

- Each instance of SUGARCREPE++ dataset consists of an image associated with three captions, where two captions describe the image and one caption does not.

Q9 **Is there a label or target associated with each instance?** *If so, please provide a description.*

- Each instance in SUGARCREPE++ consists of an image and a triplet of captions. The label for a instance is whether each caption in the triplet correctly corresponds to the image or not.

Q10 **Is any information missing from individual instances?** *If so, please provide a description, explaining why this information is missing (e.g., because it was unavailable). This does not include intentionally removed information, but might include, e.g., redacted text.*

- No.

Q11 **Are relationships between individual instances made explicit (e.g., users' movie ratings, social network links)?** *If so, please describe how these relationships are made explicit.*

- To the best of our knowledge, there is no explicit relationship between the individual instances.

Q12 **Are there recommended data splits (e.g., training, development/validation, testing)?** *If so, please provide a description of these splits, explaining the rationale behind them.*

- No, this is only an evaluation dataset.

Q13 **Are there any errors, sources of noise, or redundancies in the dataset?** *If so, please provide a description.*

- No, to the best of our knowledge there are no errors in SUGARCREPE++ dataset. We have done human validation as described in detail in the paper, to minimize any potential errors.

Q14 **Is the dataset self-contained, or does it link to or otherwise rely on external resources (e.g., websites, tweets, other datasets)?** *If it links to or relies on external resources, a) are there guarantees that they will exist, and remain constant, over time; b) are there official archival versions of the complete dataset (i.e., including the external resources as they existed at the time the dataset was created); c) are there any restrictions (e.g., licenses, fees) associated with any of the external resources that might apply to a future user? Please provide descriptions of all external resources and any restrictions associated with them, as well as links or other access points, as appropriate.*

- The images used in our dataset are based on the MS-COCO [49] dataset, which is freely and publicly available. MS-COCO dataset is released under the Creative Commons Attribution 4.0 license as listed in their website https://cocodataset.org/#termsofuse.

Q15 **Does the dataset contain data that might be considered confidential (e.g., data that is protected by legal privilege or by doctor–patient confidentiality, data that includes the content of individuals' non-public communications)?** *If so, please provide a description.*

- No, we source part of our dataset, such as image-caption pairs from MS-COCO [49] and negative captions from SUGARCREPE [29], both of which are open-source datasets.

Q16 **Does the dataset contain data that, if viewed directly, might be offensive, insulting, threatening, or might otherwise cause anxiety?** *If so, please describe why.*

- The authors did not create any content to be explicitly offensive. However, there may be instances that some users may find offensive. Since our SUGARCREPE++ dataset depends on the MS-COCO [49] and SUGARCREPE [29], we encourage the reader to refer to these datasets documentation for further details.

Q17 **Does the dataset relate to people?** *If not, you may skip the remaining questions in this section.*

- No, the dataset does not relate to people, and is not focused on people (although people may appear in the images and descriptions).

Q18 **Does the dataset identify any subpopulations (e.g., by age, gender)?**

- We explicitly do not identify any sub-populations.

Q19 **Is it possible to identify individuals (i.e., one or more natural persons), either directly or indirectly (i.e., in combination with other data) from the dataset?** *If so, please describe how.*

- Some images might contain identifiable individual faces.

Q20 **Does the dataset contain data that might be considered sensitive in any way (e.g., data that reveals racial or ethnic origins, sexual orientations, religious beliefs, political opinions or union memberships, or locations; financial or health data; biometric or genetic data; forms of government identification, such as social security numbers; criminal history)?** *If so, please provide a description.*

- We do not provide any such data in our dataset that may be considered sensitive. All images in our datasets are taken from publicly available datasets.

Q21 **Any other comments?**

- No

## L.3 Collection Process

Q22 **How was the data associated with each instance acquired?** *Was the data directly observable (e.g., raw text, movie ratings), reported by subjects (e.g., survey responses), or indirectly inferred/derived from other data (e.g., part-of-speech tags, model-based guesses for age or language)? If data was reported by subjects or indirectly inferred/derived from other data, was the data validated/verified? If so, please describe how.*

- The data associated with each instance was acquired via our data generation process (see Section 2 in our paper for a detailed description).

Q23 **What mechanisms or procedures were used to collect the data (e.g., hardware apparatus or sensor, manual human curation, software program, software API)?** *How were these mechanisms or procedures validated?*

- Please see Section 2 of our paper for a complete description of our data generation and extensive validation process.

Q24 **If the dataset is a sample from a larger set, what was the sampling strategy (e.g., deterministic, probabilistic with specific sampling probabilities)?**

- Not applicable.

Q25 **Who was involved in the data collection process (e.g., students, crowdworkers, contractors) and how were they compensated (e.g., how much were crowdworkers paid)?**

- The authors of this paper generated the textual content using generative AI as explained in Section 2 of the paper, and manually validated it.

Q26 **Over what timeframe was the data collected? Does this timeframe match the creation timeframe of the data associated with the instances (e.g., recent crawl of old news articles)?** *If not, please describe the timeframe in which the data associated with the instances was created.*

- The data was generated and evaluated over the course of approximately four months.

Q27 **Were any ethical review processes conducted (e.g., by an institutional review board)?** *If so, please provide a description of these review processes, including the outcomes, as well as a link or other access point to any supporting documentation.*

- We corresponded with the Research Ethics Board (REB) at Dalhousie University. After describing our project in detail, the REB confirmed that our project did not require ethics approval as it did not meet the regulatory definition of human subjects research. Therefore, we did not need to submit a formal application and were allowed to proceed with our research without additional REB review.

Q28 **Does the dataset relate to people?** *If not, you may skip the remaining questions in this section.*

- No, the dataset does not relate to people, and is not focused on people (although people may appear in the images and descriptions).

Q29 **Did you collect the data from the individuals in question directly, or obtain it via third parties or other sources (e.g., websites)?**

- Not applicable.

Q30 **Were the individuals in question notified about the data collection?** *If so, please describe (or show with screenshots or other information) how notice was provided, and provide a link or other access point to, or otherwise reproduce, the exact language of the notification itself.*

- Not applicable.

Q31 **Did the individuals in question consent to the collection and use of their data?** *If so, please describe (or show with screenshots or other information) how consent was requested and provided, and provide a link or other access point to, or otherwise reproduce, the exact language to which the individuals consented.*

- Not applicable.

Q32 **If consent was obtained, were the consenting individuals provided with a mechanism to revoke their consent in the future or for certain uses?** *If so, please provide a description, as well as a link or other access point to the mechanism (if appropriate).*

- Not applicable.

Q33 **Has an analysis of the potential impact of the dataset and its use on data subjects (e.g., a data protection impact analysis) been conducted?** *If so, please provide a description of this analysis, including the outcomes, as well as a link or other access point to any supporting documentation.*

- Not applicable.

Q34 **Any other comments?**

- No.

## L.4 Preprocessing, Cleaning, and/or Labeling

Q35 **Was any preprocessing/cleaning/labeling of the data done (e.g., discretization or bucketing, tokenization, part-of-speech tagging, SIFT feature extraction, removal of instances, processing of missing values)?** *If so, please provide a description. If not, you may skip the remainder of the questions in this section.*

- No preprocessing or labelling was done for creating the scenarios.

Q36 **Was the "raw" data saved in addition to the preprocessed/cleaned/labeled data (e.g., to support unanticipated future uses)?** *If so, please provide a link or other access point to the "raw" data.*

- N/A.

Q37 **Is the software used to preprocess/clean/label the instances available?** *If so, please provide a link or other access point.*

- Not applicable

Q38 **Any other comments?**

-

## L.5 Uses

Q39 **Has the dataset been used for any tasks already?** *If so, please provide a description.*

- No. SUGARCREPE++ is a new benchmark.

Q40 **Is there a repository that links to any or all papers or systems that use the dataset?** *If so, please provide a link or other access point.*

- To the best of our ability, we will try to maintain links to derivative papers and systems that use our dataset in the SUGARCREPE++ GitHub repository (https://github.com/Sri-Harsha/scpp).

Q41 **What (other) tasks could the dataset be used for?**

- The primary use case of our benchmark is to evaluate the sensitivity of VLMs and ULMs to semantic and lexical alterations. While we did not explore this direction in the present work, future work can use this dataset to evaluate any multi-modal system that uses VLMs and ULMs as foundation blocks such as text-to-image retrieval models, multi-modal chatbots, etc.

Q42 **Is there anything about the composition of the dataset or the way it was collected and preprocessed/cleaned/labeled that might impact future uses?** *For example, is there anything that a future user might need to know to avoid uses that could result in unfair treatment of individuals or groups (e.g., stereotyping, quality of service issues) or other undesirable harms (e.g., financial harms, legal risks) If so, please provide a description. Is there anything a future user could do to mitigate these undesirable harms?*

- Due to the reliance on the MS-COCO [49] and SUGARCREPE [29] datasets, SUGARCREPE++ may contain offensive material, or biases present in these source datasets. Users of SUGARCREPE++ should carefully consider how these limitations may impact their potential use case and exercise discretion in their application of the dataset.

Q43 **Are there tasks for which the dataset should not be used?** *If so, please provide a description.*

- The dataset should be avoided for a task if the limitations discussed above are unacceptable or potentially problematic for the intended use case.

Q44 **Any other comments?**

- No.

## L.6 Distribution and License

Q45 **Will the dataset be distributed to third parties outside of the entity (e.g., company, institution, organization) on behalf of which the dataset was created?** *If so, please provide a description.*

- Yes, SUGARCREPE++ dataset will be open-sourced and freely available.

Q46 **How will the dataset be distributed (e.g., tarball on website, API, GitHub)?** *Does the dataset have a digital object identifier (DOI)?*

- Our datset and code will be made available at the following Github link: https://github.com/Sri-Harsha/scpp

Q47 **When will the dataset be distributed?**

- October 31, 2024 and onward.

Q48 **Will the dataset be distributed under a copyright or other intellectual property (IP) license, and/or under applicable terms of use (ToU)?** *If so, please describe this license and/or ToU, and provide a link or other access point to, or otherwise reproduce, any relevant licensing terms or ToU, as well as any fees associated with these restrictions.*

- We release data under the **CC-BY-4.0** license.
- Our code will be released under the **Apache-2.0** license

Q49 **Have any third parties imposed IP-based or other restrictions on the data associated with the instances?** *If so, please describe these restrictions, and provide a link or other access point to, or otherwise reproduce, any relevant licensing terms, as well as any fees associated with these restrictions.*

- The dataset will be released under CC-BY-4.0 license.

Q50 **Do any export controls or other regulatory restrictions apply to the dataset or to individual instances?** *If so, please describe these restrictions, and provide a link or other access point to, or otherwise reproduce, any supporting documentation.*

- No.

Q51 **Any other comments?**

- No.

### L.7 Maintenance

**Q52 Who will be supporting/hosting/maintaining the dataset?**

- The authors will be supporting, hosting and maintaining the dataset and code through GitHub.

**Q53 How can the owner/curator/manager of the dataset be contacted (e.g., email address)?**

- The authors can be contacted through their email. Alternatively, an issue can be created on our GitHub repository.

**Q54 Is there an erratum?** *If so, please provide a link or other access point.*

- There is no erratum for our initial release. Errata will be documented as future releases on the benchmark website.

**Q55 Will the dataset be updated (e.g., to correct labeling errors, add new instances, delete instances)?** *If so, please describe how often, by whom, and how updates will be communicated to users (e.g., mailing list, GitHub)?*

- SUGARCREPE++ will be updated. Updates can be monitored through Github.

**Q56 If the dataset relates to people, are there applicable limits on the retention of the data associated with the instances (e.g., were individuals in question told that their data would be retained for a fixed period of time and then deleted)?** *If so, please describe these limits and explain how they will be enforced.*

- NA

**Q57 Will older versions of the dataset continue to be supported/hosted/maintained?** *If so, please describe how. If not, please describe how its obsolescence will be communicated to users.*

- We will host older versions in GitHub, in case we release newer versions.

**Q58 If others want to extend/augment/build on/contribute to the dataset, is there a mechanism for them to do so?** *If so, please provide a description. Will these contributions be validated/verified? If so, please describe how. If not, why not? Is there a process for communicating/distributing these contributions to other users? If so, please provide a description.*

- Users can extend and build on SUGARCREPE++ dataset as we did for SUGARCREPE. We do not take responsibility for validating any extension of our work.

**Q59 Any other comments?**

- No.

