# OpenReview forum: "SUGARCREPE++ Dataset: Vision-Language Model Sensitivity to Semantic and Lexical Alterations"
_NeurIPS.cc/2024/Datasets_and_Benchmarks_Track — NeurIPS 2024 Track Datasets and Benchmarks Poster_

### Official Review · Reviewer_XGk7 · 2024-06-15
**Interesting benchmark with room for improvement**

**Rating:** 6
**Confidence:** 5
**Correctness:** The collection method is well documen…

**Review:**

I believe the paper studies an interesting problem of lexical vs semantic for vision-language reasoning, however, the paper's writing and clarity can be further improved. This is the first benchmark that studies this issue so it is indeed novel. The experimental results are somewhat expected as previous work also claims text encoders are the bottleneck for vision-language models (which is okay).

**Strengths:**

While the research community now agrees that discriminative VLMs like CLIP are bags-of-words representations, it is unclear to what extent they are just confused by lexically similar captions. This benchmark is an interesting first step towards studying such biases. The paper documents the collection pipeline in detail to make it easy to reproduce the results. The dataset is hosted on HuggingFace to make it accessible to all researchers.

**Additional Feedback:**

Please use the reviewers' feedback to improve on the paper!


I increase my rating from 4 to 6 after the author's rebuttal. The biggest concern is the presentation, which the authors promise to address in the revised version.

**Clarity:**

No, I will encourage authors to improve their readability by providing clear definitions and simple examples.

**Documentation:**

The dataset URL is included in the supplement.

**Ethics:**

No.

**Limitations:**

I believe the authors can vastly strengthen this paper through better presentation. For example, some sentences (even the below one in the abstract) are too long/complicated to read:

"We show that all the models which achieve better performance on compositionality datasets need not perform equally well on SUGARCREPE++ signifying that compositionality alone may not be sufficient for understanding semantic and lexical alterations."

The paper claims that previous benchmarks use "Varying Definitions of Semantic Equivalence", but this paper does not even provide a simple and clear definition of semantic equivalence.

Lastly, the paper should discuss more recent work that uses generative VLMs to perform image-text retrieval tasks, such as VisualGPTScore [1]. In addition, [1] also studies the language biases of SugarCrepe, which allows it to be solved by blind solutions that only look at captions.

**References**

[1] Revisiting the Role of Language Priors in Vision-Language Models. Lin et al. ICML 2024.

**Opportunities For Improvement:**

My biggest concern is that the paper did not carefully define how to measure "semantic" and "lexical" variations. Although this is intuitive for most NLP researchers, I believe providing a proper definition (ideally with some numerical metrics, and they don't need to be implementable) will be helpful for readers not from a NLP background (e.g., vision researchers). For example, I found these two sentences in the dataset:

Original caption: Frisbee in the air, a dog squatting looking up at it, and a person standing behind the dog watching it, on the grass with a tree.

Rewritten caption:
The Frisbee is in the air, with a dog squatting and looking up at it, and a person standing behind the dog, observing it, on the grass with a tree.

For me, these two sentences look almost the same both semantically and lexically (there are only very few lexical variations).

I thought the authors were going to conclude that VLMs are simply confused by lexically similar positive and negative captions, and a naive solution is to make the two captions lexically different before doing retrieval tasks. Clearly, this is not the conclusion of the paper, as it seems VLMs are still confused by two lexically different and semantically different captions (indicated in Table 6). I am not sure when to what extent the conclusion diverges from the original SugarCrepe paper.

**Relation To Prior Work:**

Yes.

**Summary And Contributions:**

The paper proposes to augment SugarCrepe, an existing image-to-text retrieval benchmark, with additional positive captions that have significant lexical variation (but not semantic variation). The authors use ChatGPT to rewrite the original positive caption while using some heuristic rules to avoid generating duplicate or superfluous captions. The authors then evaluate this benchmark and claim that existing VLMs (and LLMs) tend to incorrectly match lexically similar captions regardless of semantic meanings.

---

> ### Author Rebuttal · Authors · 2024-08-20
>
> We thank you for your detailed review with recommendations for improving our presentation. We also thank you for recognizing the uniqueness and importance of our evaluation technique with the new SugarCrepe++ dataset. In the following response, we aim to address your concerns.
>
> ***(Q1) My biggest concern is that the paper did not carefully define how to measure "semantic" and "lexical" variations. [..] I found these two sentences in the dataset:***
>
> >***Original caption: Frisbee in the air, a dog squatting looking up at it, and a person standing behind the dog watching it, on the grass with a tree.***
>
> >***Rewritten caption: The Frisbee is in the air, with a dog squatting and looking up at it, and a person standing behind the dog, observing it, on the grass with a tree.***
>
> ***For me, these two sentences look almost the same both semantically and lexically (there are only very few lexical variations).***
>
> Great suggestion! As per your suggestion, we provided definitions and described a metric for measuring lexical/syntactic similarity (please see the global response). We apply this metric to analyze caption pairs in SugarCrepe++ and visualize the statistics in Figure XGk7.1 showing that {P1,P2} have lower lexical/syntactic similarity as compared to {P1,N}.
>
> We sincerely appreciate the reviewer’s dedication in not only reviewing our paper but also taking the time to examine our dataset on HuggingFace! For the given pair of sentences, the LexicalSimilarity is ~0.85 while the SyntacticSimilarity is ~0.30 resulting in an SLS score of ~0.26. The smaller syntactic similarity indicates that the two sentences vary in terms of their constituency parse trees although they look very similar superficially (0.85 lexical-similarity).
>
> ***
>
> ***(Q2) I thought the authors were going to conclude that [..] and a naive solution is to make the two captions lexically different before doing retrieval tasks. Clearly, this is not the conclusion of the paper [..](indicated in Table 6). I am not sure when to what extent the conclusion diverges from the original SugarCrepe paper.***
>
> Thank you for raising this point. We emphasize that the goal of our paper is to introduce a new dataset for evaluating language models (both VLMs as well as unimodal language models) in terms of their semantic understanding abilities beyond the lexical and syntactic forms. We believe there may have been a slight misunderstanding regarding Table 6. In the following, we will first describe Table 6 and then discuss our conclusions.
>
> In Table 6, the columns corresponding to SugarCrepe++(SC++) evaluate the ability of VLMs to identify the negative caption out of {P1, P2, N} whereas the columns corresponding to SugarCrepe (SC) evaluate the ability of VLMs to identify the negative caption out of {P1, N}. In contrast to SC, SC++ is a harder task focused on evaluating fundamental language understanding. By performing a side-by-side comparison between SC and SC++, Table 6 demonstrates that models that perform better on SC do not necessarily perform well on SC++. For instance, ALBEF and XVLM consistently outperform CLIP on SC across all subsets (e.g., swap-object, swap-attribute, etc), yet consistently score lower than CLIP on SC++ in every subset.
>
> The specific setup described by the reviewer refers to the task of selecting the incorrect caption between a pair of lexically different sentences (i.e., {P2, N}). This is a new setting different from both SC and SC++ – we will call this setting as P2vsN. We collected results on P2vsN and summarized them in Table XGk7.2. For reference, we also include the corresponding columns from Table 6. While we observe that P2vsN is easier for some models in some cases, on average P2vsN is still challenging for most VLMs.
>
> Overall, our experiments with lexical/syntactic variations show that VLMs struggle to encode the correct semantics. While SugarCrepe focuses on evaluating compositional reasoning in language models, our conclusions point towards a fundamental problem in language understanding (both semantic as well as syntactic/lexical aspects). More importantly, we extend this evaluation to unimodal language models and surprisingly observe similar patterns showing that unimodal language models also face similar challenges in identifying semantic equivalences beyond the lexical representations.
>
> ***
>
> ***(Q3)  I believe the authors can vastly strengthen this paper through better presentation. For example, some sentences (even the below one in the abstract) are too long/complicated to read: [..]***
>
> We thank the reviewer for this suggestion. We will identify and rewrite long sentences for improved readability in our next revision.
>
> ***
>
> ***(Q4) The paper claims that previous benchmarks use "Varying Definitions of Semantic Equivalence", but this paper does not even provide a simple and clear definition of semantic equivalence.***
>
> We agree that defining semantic equivalence is important, and so in page 2 of our original submission, we explained semantic equivalence as follows:
>
> “Two sentences are said to be semantically equivalent if the sentences convey the same meaning and can be inferred from each other (i.e., bidirectional entailment).”
>
> We then provided additional discussion on definitions in the literature.
>
> We intended and believe this definition to be clear for non-NLP readers, but if this reviewer could identify any points of confusion, we will be very happy to clarify.  Also, if the reviewer feels it would be helpful, we could put this explanation in italics to emphasize it in the final paper.
>
> ***

---

> > ### Author Rebuttal · Authors · 2024-08-20
> >
> > ***(Q5) Lastly, the paper should discuss more recent work that uses generative VLMs to perform image-text retrieval tasks, such as VisualGPTScore [1].***
> >
> > We thank the reviewer for bringing these references to our attention. Following the reviewer’s suggestion, we consider five additional generative VLMs and evaluate two recent methods — VisualGPTScore [1] and VQAScore [2] — on the SugarCrepe++ dataset. For reference, we also include the corresponding results on SugarCrepe (See Table XGk7.1). Both VisualGPTScore and VQAScore are methods designed to make the most effective use of generative VLMs for image-text matching tasks.
> >
> > We further note that the additionally considered generative VLMs range between 3B-11B in parameter counts and are larger than most VLMs previously considered in the paper — from Tables 8 and 9, we can see that the largest model we considered is a 2.5B parameter CLIP model (ViT-bigG/14). Despite using significantly larger models, we find that VisualGPTScore still performs comparably with most other models already included in the paper (e.g., Table 2). However, using the same generative VLMs, VQAScore achieves significantly improved performance. Nevertheless, we note that this performance is comparable with that of GPT-4o (Table 1 of Supplementary PDF) and in general, a significant gap still exists between the current performance and human-level performance.
> >
> > We feel that these new evaluations, prompted by your review, strengthen our contribution and will include it in the final version.
> > ***
> >
> > ***(Q6) In addition, [1] also studies the language biases of SugarCrepe, which allows it to be solved by blind solutions that only look at captions.***
> >
> > Thank you for raising this point. Unlike previous datasets such as ARO and CREPE, blind inference of generative VLMs — using Gaussian images — on SugarCrepe as well as SugarCrepe++ does not exceed the performance obtained with the use of images. Thus, we believe that blind solutions exploiting language biases *solve* neither the SugarCrepe nor the  SugarCrepe++ datasets.
> >
> > We take this opportunity to highlight that SugarCrepe++ is designed such that the 3 captions {P1, P2, N} can be analyzed independent of the image to select the semantically different sentence. This can be interpreted as a variant of blind evaluation since the model is not allowed to use the image as input. Different from previous approaches to blind evaluation that select the incorrect caption solely based on language bias (e.g., blind-VisualGPT Score, Grammar Score, VERA score), our text-only task (TOT) setup allows the model to encode and compare between the three sentences in order to select the semantically different sentence (i.e., N). Thus, our approach to blind evaluation is ultimately an evaluation of language understanding (e.g., see Table 7 and TOT columns in Tables 2, 3, 4 & 5) rather than language bias.
> >
> > In conclusion, SugarCrepe++ is a challenging dataset for evaluating language understanding and offers valuable insights into the model performance.
> >
> > [1] Revisiting the Role of Language Priors in Vision-Language Models. ICML 2024.
> >
> > [2] Evaluating Text-to-Visual Generation with Image-to-Text Generation. ECCV 2024.
> > ***
> >
> > We hope we have addressed all your concerns and look forward to discussing any outstanding concerns in the remaining rebuttal discussion period.

---

> > ### Comment · Reviewer_XGk7 · 2024-08-20
> > **Thanks for the rebuttal**
> >
> > Thank you for the detailed rebuttal. I have decided to increase my ratings from 4 to 5 (still not final) and I have some follow-up questions/suggestions:
> >
> > Q1: I am still surprised that these two sentences result in a low SyntacticSimilarity score. As a reader without an NLP background, I would appreciate it if the authors could provide (1) more example caption pairs (in both rebuttal and paper) with their LexicalSimilarity and SyntacticSimilarity scores and (2) for this particular example, could you explain why these two sentences result in such a low SyntacticSimilarity score (e.g., by providing another caption that looks vastly different but has a higher SyntacticSimilarity score)?
> >
> > Q2: Is SC++ a 3-choose-1 task with a random chance accuracy of 33.3%? If so, Table 6 might be misleading in comparing SC and SC++. I think it is important to provide the random chance accuracy in this table for reference. I appreciate the new P2vsN experiment.
> >
> > Q3 and Q4: Thank you for the clarification. I believe my confusion comes from the complex and long descriptions of prior work from this same paragraph in the introduction section, for example, *"Different from semantically equivalent sentences, a pair of questions are defined to be semantically equivalent if they have the same answer [57] and hence, datasets on semantically equivalent question pairs (e.g., QQP) require additional knowledge beyond language understanding."* I don't see how discussing a (not-so-relevant) QA dataset helps non-NLP readers appreciate this concept of semantic equivalence. Overall, I believe a better presentation (simpler and more concise sentences, ideally with simple and concrete examples to illustrate these NLP concepts) can lead to a much stronger paper.

---

> > > ### Comment · Reviewer_XGk7 · 2024-08-20
> > > **What is the conclusion for generative VLMs?**
> > >
> > > Lastly, the new experiments using VGPTScore and VQAScore are very interesting! However, I don't see any discussion about this experiment in the rebuttal.
> > >
> > > I would appreciate it if the authors could highlight the conclusion of new VGPTScore/VQAScore experiments with a summary (e.g., how generative VLMs perform against the discriminative VLMs) and describe how these new experiments add to the paper (e.g., are generative VLMs better at semantic equivalence against lexical/syntactic variations).

---

> > > > ### Author Response · Authors · 2024-08-23
> > > > **Thank you for your response!**
> > > >
> > > > We thank you for your prompt response and giving us an opportunity to address your remaining concerns. We thank you for raising your score and pleased to know that some of your concerns were addressed. In the response below, we aim to resolve the remaining issues.
> > > >
> > > > ***
> > > >
> > > > > ***I am still surprised that [..] low SyntacticSimilarity score.***
> > > >
> > > > These results can indeed seem surprising at first glance, and we’re glad to explain why this might be the case. For clarity, we will summarize the lexical-similarity and syntactic-similarity measures described in the global response:
> > > >
> > > > - **Lexical-Similarity**: By definition, lexical refers to words/vocabulary and hence, lexical-similarity compares a pair of sentences at the word-level. In particular, a higher overlap of vocabulary and order of occurrence should lead to higher lexical similarity. We measure lexical-similarity using normalized Levenshtein similarity.
> > > >
> > > > - **Syntactic-Similarity**: Syntax refers to the arrangement of words to form constituents (e.g., phrase/clause) and hence, syntactic-similarity compares a pair of sentences based on their constituency parse trees. Thus, syntactic similarity is higher for sentences whose constituency parse trees share a common structure. We use the FastKASSIM score to measure syntactic-similarity.
> > > >
> > > > Lexical similarity is more aligned with human perception as compared to syntactic-similarity but does not consider grammatical nuances while estimating similarity. In your example, the two sentences have several shared words occurring in the same sequence:
> > > > > Frisbee in the air, a dog squatting looking up at it, and a person standing behind the dog watching it, on the grass with a tree.
> > > >
> > > > > The Frisbee is in the air, with a dog squatting and looking up at it, and a person standing behind the dog, observing it, on the grass with a tree.
> > > >
> > > > Thus, we observe a high lexical similarity of ~0.85 and this correlates with our perceived sense of similarity between the two sentences.
> > > >
> > > > On the other hand, constituency parse trees are constructed using rules of grammar and seemingly minor changes to a sentence can introduce important structural changes in the parse-tree. The above pair of sentences best demonstrate this: despite high lexical similarity, we observe a lower syntactic-similarity score of ~0.3.
> > > > ***
> > > > Further, we explain syntactic-similarity with a simpler example (not from our dataset) by considering:
> > > >
> > > > Sentence-1: A boy sits.
> > > >
> > > > Sentence-2: The girl runs.
> > > >
> > > > The above two sentences are semantically different i.e., they convey very different meanings but the syntactic structure of these two sentences is identical as shown below :
> > > >
> > > > ||||
> > > > |:---:|:---:|:---:|
> > > > |A|boy|sits|
> > > > |(Article)|(Noun)|(Verb)|
> > > > |The|girl|runs|
> > > >
> > > > Accordingly, the FastKASSIM syntactic-similarity score between these two sentences is 1.0. – This shows that two sentences with similar (same) syntactic structure will have a higher (nearly 1.0) syntactic similarity score.
> > > >
> > > > We hope that our clarifications were helpful and refer to FastKASSIM [1] for more details on the internal workings of their algorithm.
> > > >
> > > > [1] FastKASSIM: A Fast Tree Kernel-Based Syntactic Similarity Metric. EACL 2023.
> > > > ***
> > > >
> > > > > ***more example caption pairs (in both rebuttal and paper) with their LexicalSimilarity and SyntacticSimilarity scores***
> > > >
> > > > As per your suggestion, we provide examples of sentence-pairs from our dataset along with their syntactic-similarity and lexical-similarity scores. We will expand upon this table in the final version.
> > > >
> > > > | Sentence 1 | Sentence 2 | Syntactic Similarity | Lexical Similarity | SLS |
> > > > |---|---|---|---|---|
> > > > | A living room with white furniture and a small wooden table. | A living room without white furniture and a small wooden table. | 1.00 | 0.95 | 0.95 |
> > > > | Three adult zebras walk calmly along close together. | Three adult horses walk calmly along close together. | 1.00 | 0.91 | 0.91 |
> > > > | A street light in front of a colorful train on a bridge. | A colorful train is on a bridge with a street light in front of it. | 0.84 | 0.28 | 0.24 |
> > > > | A teddy bear is placed on a metallic sculpture. | The metallic sculpture is positioned below the teddy bear. | 1.00 | 0.24 | 0.24 |
> > > > | A desk with two computer monitors, two mice, a cup and a keyboard on it. | There are two computer monitors, two mice, a cup, and a keyboard on the desk. | 0.26 | 0.76 | 0.20 |
> > > > | A fire hydrant is decorated with an American flag design. | The American flag design is adorned on the fire hydrant. | 0.96 | 0.19 | 0.18 |
> > > > | An empty clean kitchen with cabinetry, stove and dishwasher. | An empty kitchen featuring cabinets, stove, and a dishwasher is clean. | 0.30 | 0.60 | 0.18 |
> > > > | A table topped with apples, oranges and bananas. | The table stands as a backdrop to a fruitful display, showing apples, oranges, and bananas arranged on top. | 0.11 | 0.40 | 0.04 |
> > > > | A white chair, books and shelves and a tv on in this room. | In this room, there is a white chair, shelves, books, and a TV on. | 0.12 | 0.27 | 0.03 |

---

> > > > > ### Author Response · Authors · 2024-08-23
> > > > > **Thank you for your response! (contd)**
> > > > >
> > > > > > ***for this particular example, could you explain why these two sentences result in such a low SyntacticSimilarity score (e.g., by providing another caption that looks vastly different but has a higher SyntacticSimilarity score)?***
> > > > >
> > > > > As discussed above, these two sentences have a low syntactic-similarity score due to important differences between their parse-trees. While the sentence pair in your example is from the replace-attribute subset, we discovered another example with the same Original-caption in the replace-relation subset. For clarity, we summarize the captions below:
> > > > >
> > > > > > (a) Original Caption: Frisbee in the air, a dog squatting looking up at it, and a person standing behind the dog watching it, on the grass with a tree.
> > > > >
> > > > > > (b) Re-written Caption (Replace-Attribute): The Frisbee is in the air, with a dog squatting and looking up at it, and a person standing behind the dog, observing it, on the grass with a tree.
> > > > >
> > > > > > (c) Re-written Caption (Replace-Relation); The person is standing behind the dog as it watches the Frisbee in mid-air, on the grass with a nearby tree.
> > > > >
> > > > > > (d) Negative Caption (Replace-Attribute): Frisbee in the air, a dog squatting looking up at it, and a person sitting behind the dog watching it, on the grass with a tree.
> > > > >
> > > > > > (e) Negative Caption (Replace-Relation): Frisbee in the air, a dog jumping and looking up at it, and a person standing behind the dog watching it, on the grass with a tree.
> > > > >
> > > > > While (a) and (b) have syntactic-similarity of ~0.30, (a) and (c) have a syntactic similarity of ~0.59. Similarly, (a) and (b) have a lexical-similarity of ~0.84 while (a) and (c) have a lexical-similarity of ~0.37.
> > > > > Together, (a) and (b) have an SLS score of ~0.25 while (a) and (c) have a SLS score of ~0.22.
> > > > >
> > > > > Here, (c) is indeed an example that looks vastly different from (a) while having a greater syntactic similarity score.
> > > > >
> > > > > To understand how models perform on these examples, we evaluated 25 VLMs in the TOT (text-only task) setting. Interestingly, we found that none of the models were able to successfully identify the negative example in both of these cases! This demonstrates that both examples are challenging for several models.
> > > > >
> > > > > ***
> > > > > > ***Is SC++ a 3-choose-1 [...] I appreciate the new P2vsN experiment.***
> > > > >
> > > > > Thank you for suggesting this! Yes, SC++ has a random chance accuracy of 33.3% while SC has a random chance accuracy of 50% — we will indicate this. Since SC and SC++ accuracies are not directly comparable, we compare the relative ordering of models. For example, Table 6 shows that models performing better than CLIP on SC may not behave similarly on SC++.
> > > > >
> > > > > In Table 6 of main paper, we use CLIP’s performance as a reference to compare the trends in model performance and make the following observations:
> > > > >
> > > > > - ALBEF and XVLM models consistently perform better than CLIP on SC across all subsets. However, both ALBEF and XVLM show a degradation in performance compared to CLIP on SC++.
> > > > >
> > > > > - While BLIP and NegCLIP perform better than CLIP on both SC and SC++ across four subsets, we observe a trend reversal on the replace relation subset. In particular, BLIP > CLIP > NegCLIP on SC++ but BLIP < CLIP < NegCLIP on SC.
> > > > >
> > > > > From these observations, we can infer that a model’s SugarCrepe performance may not be predictive of its SugarCrepe++ performance highlighting the distinctiveness of SugarCrepe++.
> > > > > ***
> > > > > > ***Thank you for the clarification. [...] much stronger paper.***
> > > > >
> > > > > We value your suggestions for improved presentation. We feel it is important to discuss previous works that share a similar goal in order to position our new dataset and highlight its distinctive properties with respect to previous works. In particular, our goal was to provide the reader a perspective of how semantic equivalence is being discussed and addressed in the literature. However, we now understand your specific concerns regarding our presentation of semantic equivalence. So, we will reorganize parts of the introduction to avoid distracting the reader from the main point while also ensuring that we acknowledge previous works.
> > > > >
> > > > > We will indeed fix the presentation issues. In fact, we carefully reviewed our submitted version and already identified and revised the few presentation issues in our draft including the two mentioned by you.
> > > > >
> > > > > For instance, we have rewritten the statement
> > > > >
> > > > > > We show that all the models which achieve better performance on compositionality datasets need not perform equally well on SUGARCREPE++ signifying that compositionality alone may not be sufficient for understanding semantic and lexical alterations.
> > > > >
> > > > > as
> > > > >
> > > > > > We demonstrate that models excelling on compositionality datasets may not perform as well on SUGARCREPE++. This indicates that compositionality alone might not be sufficient to fully understand semantic and lexical alterations.

---

> > > > > > ### Author Response · Authors · 2024-08-23
> > > > > > **Thank you for your response! (contd.)**
> > > > > >
> > > > > > Similarly,
> > > > > > > Earlier version: “Recognizing semantic similarity is often viewed as being fundamental to language understanding, and strong performance on semantic text similarity is often predictive of a language model’s performance in various downstream applications including question-answering, retrieval and summarization.”
> > > > > >
> > > > > > > Current version: “Understanding semantic similarity is key to language comprehension. Good performance in semantic text similarity usually predicts how well a language model will perform in downstream tasks such as question-answering, retrieval, and summarization.”
> > > > > >
> > > > > > Your suggestions are incredibly helpful and sincerely appreciated — we look forward to presenting a refined, clear and well-written version of this work in our final paper.
> > > > > >
> > > > > > ***
> > > > > >
> > > > > > > ***Lastly, the new experiments using VGPTScore and VQAScore are very interesting! [...] lexical/syntactic variations).***
> > > > > >
> > > > > > We agree with the reviewer that the new experimental results with VGPTScore and VQAScore are indeed interesting! We would like to clarify that we presented our conclusions regarding these new experiments in Q5 of our rebuttal response above. We highlight the conclusions as follows:
> > > > > >
> > > > > > 1) Despite using significantly larger models (3B-11B parameters), VGPTScore performs comparably with several discriminative VLMs previously considered in our paper.
> > > > > >
> > > > > > 2) Interestingly, using the same generative VLMs, VQAScore achieves significant performance improvements on SC++ as compared to VGPTScore. However, this is still below human performance signifying opportunity for further improvement.
> > > > > >
> > > > > > We take this opportunity to share some of our other experiments with generative VLMs:
> > > > > >
> > > > > > - **GPT-4o Experiments**: we experimented with three different prompts that are equivalent and observed differences in performance likely due to their prompt sensitivity. We discuss this in Section 1.1 of our supplementary PDF.
> > > > > >
> > > > > > - **BLIP/BakLLaVA Experiments**: We repeated the above experiments with BLIP and BakLLaVA and identified similar behavior. We discuss these experiments in our rebuttal response to reviewer DXNt (Q3) and provide the results in Tables DXNt.3 and DXNt.4.
> > > > > >
> > > > > > Based on several experiments with generative-VLMs, we can conclude that the performance of generative-VLMs ultimately depends on the inference-technique (e.g., VGPTScore/VQAScore/Prompting-styles) and with the right inference-technique, generative-VLMs may outperform discriminative models. We will include these new experiments in the final version of our paper and expand upon the discussion included in our rebuttal responses.
> > > > > >
> > > > > > ***
> > > > > >
> > > > > > We hope we have addressed all your concerns and look forward to discussing any outstanding concerns in the remaining rebuttal discussion period.

---

> > > > > > > ### Comment · Reviewer_XGk7 · 2024-08-23
> > > > > > >
> > > > > > > Thank you for the detailed rebuttal, which is very helpful. The generative VLMs experiments are interesting and their conclusions can imply interesting follow-up works.

---

> > > > > > > > ### Author Response · Authors · 2024-08-24
> > > > > > > > **Thank you**
> > > > > > > >
> > > > > > > > Thank you for considering our rebuttal, and for your active engagement throughout the entire rebuttal period. We appreciate your detailed feedback; it has been invaluable in improving the clarity and presentation of our manuscript.

---

### Official Review · Reviewer_a1nX · 2024-07-24
**A benchmark on model consistency over semantically equivalent captions with good analysis**

**Rating:** 7
**Confidence:** 4
**Correctness:** The claims seem to be correct.
**Clarity:** Yes

**Review:**

**Pros**:
- The paper is well-written and easy to follow, with the research question well-motivated;
- The proposed benchmark is novel and poses a central challenge of understanding semantic equivalence to VLMs;
- The analysis is relatively comprehensive, including evaluations on various VLMs and explanations of how different components of the model (architectures, objectives, training data) results in different performance on the proposed benchmark;
- The analysis also compares the evaluation of VLMs on SUGARCREPE++ with the evaluation on the previous SUGARCREPE, highlighting the main contribution of the proposed benchmark;

**Cons**:
- The research question motivated in this paper is a unimodal problem in language understanding, while the necessity of addressing such a language understanding issue in multimodal setting (as the evaluation focus is on VLMs) was not properly highlighted;
- The claim that "Text encoders bottleneck VLM performance on SUGARCREPE++" seems a little trivial as the benchmark is designed to evaluate model's language understanding of semantically equivalent texts and with the additional visual information, the performance is expected to be better

**Strengths:**

Overall, the paper addresses current models' incapability in distinguishing semantic and lexical alternations in text, which remains an open problem for model consistency in LLM development. The paper has a relatively comprehensive documentation of the data collection process and appropriate justification for the procedures. The following analysis has exposed a general lack of understanding of semantic equivalence in existing language models. Comparison between various models on different subsets of the benchmark provides insights to which component of architecture, objectives, and data contributes to different failure patterns.

**Additional Feedback:**

See above (Opportunities For Improvement)

**Documentation:**

Yes

**Limitations:**

Did not find the limitation section or in the discussion.

**Opportunities For Improvement:**

The discussion should highlights the necessity of improving text encoders on such a semantic equivalence task by connecting this language understanding issue with the observed significantly degraded performance in multimodal tasks. There also exists a wide range of work targeting at addressing the model consistency issue. Including evaluation of these methods on the benchmark will be meaningful for comparison.

**Relation To Prior Work:**

Yes

**Summary And Contributions:**

This paper proposed a benchmark SUGARCREPE++ on evaluating language understanding of semantic equivalent expressions for multimodal and unimodal language models. Building upon the original SUGARCREPE, which is designed to test model's ability in distinguishing semantically different but lexically similar captions, the proposed benchmark collects one additional caption semantically equivalent to the ground truth caption. Evaluation on this benchmark exposes significant problem of existing models in understanding semantic and lexical variations and attributes this issue to the insufficient text encoders.

---

> ### Author Rebuttal · Authors · 2024-08-20
>
> We thank you for the thorough review with insightful suggestions for improvement. We are happy to see that you also liked our presentation and comprehensive analysis. We agree that the proposed benchmark indeed poses a central challenge of understanding semantic equivalence to VLMs. In the following response, we aim to address the weaknesses and answer your questions.
>
> ***(Q1) The research question motivated in this paper is a unimodal problem in language understanding, while the necessity of addressing such a language understanding issue in multimodal setting (as the evaluation focus is on VLMs) was not properly highlighted.***
>
> Thank you for bringing this to our attention. Our research question is focused on evaluating language understanding — specifically, the semantic understanding of sentences beyond their syntactic/lexical representations — both in VLMs (Table 2, in our main paper) and ULMs (Tables 7 and 11, in our main paper). We appreciate your suggestion and we will ensure that this is clearly highlighted in our final revision.
> ***
> ***(Q2) The claim that "Text encoders bottleneck VLM performance on SUGARCREPE++" seems a little trivial as the benchmark is designed to evaluate model's language understanding of semantically equivalent texts and with the additional visual information, the performance is expected to be better***
>
> We note that SugarCrepe++ is designed such that the incorrect caption (N) can be selected amongst the three captions {P1, P2, N} even without visual grounding – we refer to this as the Text-only Task (ToT). Ideally, we expect the VLM performance in ToT setting to be comparable to the performance in Image-to-Text (ITT) setting. However, we find that VLMs in ToT setup do not perform as well as in the ITT setup leading to our conclusion that text encoders bottleneck VLM performance on SugarCrepe++. We additionally note that we also evaluate the unimodal language models in Table 7 where we generally observe higher performance in the ToT setup: for example, Sentence-T5-xl and Angle-Llama-7b-nli-v2 which on average perform better than most of the VLMs in the text-only task (ToT). We will clarify this in the final revision.
>
> ***
>
> ***(Q2b) The discussion should highlight the necessity of improving text encoders on such a semantic equivalence task by connecting this language understanding issue with the observed significantly degraded performance in multimodal tasks.***
>
> Good point! Our evaluation with the SugarCrepe++ dataset indeed echoes the necessity of improving text encoders in terms of their semantic language understanding abilities.
>
> In our evaluation of VLMs (section 3.1), we evaluate the VLMs in the text-only task (TOT) and find that “Text encoders bottleneck VLM performance on SugarCrepe++”. We subsequently analyze previous methods for training improved VLMs and find (i) Fine-tuning VLMs for image-text retrieval improves performance with opportunity for further improvements; and (ii) Compositionality enhancing methods improve performance on SUGARCREPE++ by strengthening the VLM text encoder.
>
> Collectively, all these findings highlight the importance of improved text encoders. Following your recommendation, we will ensure that this is explicitly highlighted in our final revision.
>
> ***
>
> ***(Q3) There also exists a wide range of work targeting at addressing the model consistency issue. Including evaluation of these methods on the benchmark will be meaningful for comparison.***
>
> Good suggestion! We describe the experimental results in detail below, following a brief overview:
>
> **Overview:**  Most previous works on consistency issues have focused primarily on either text or image modalities, but not on the multi-modal vision-language setting. We adapted some approaches proposed for LLMs, such as Ensemble of models [1, 2] and ensemble of augmented prompts [3], to VLMs for evaluation on SugarCrepe++. Please note that we evaluated the consistency approaches we could think of and would greatly appreciate any specific approaches you suggest; we would be happy to evaluate those on SugarCrepe++. Additionally, we found that VLMs are highly susceptible to variations in prompts, with even small changes significantly affecting model performance on SugarCrepe++ (for more details, please refer to our response to Reviewer DXNt:Q3). This demonstrates that SugarCrepe++ can be a valuable tool for evaluating any future approaches aimed at improving VLM consistency.
>
> **Ensemble of models:** In this approach, we combine the outputs of multiple VLMs to obtain the final decision using majority voting. Table 1 compares the performance of  individual VLMs with the ensemble models. We analyzed three different ensembles:
>
> 1) Ensemble-All: all the VLMs analyzed in the paper.
>
> 2) Ensemble-1: Set of best performing variants of CLIP [RoBERTa-Vit-B/32, ViT-H/14, Vit-g/14, xlm-RoBERTa-base-ViT-B/32, xlm-RoBERTa-large-ViT-H/14] and
>
> 3) Ensemble-2: Best performing models from each sub-category of VLMs -- [FLAVA, BLIP, NegCLIP, ViLT-ITR-COCO, ViT-bigG/14]
>
> Table a1nX.1 presents the performance of the three different ensemble models. Both Ensemble-1 and Ensemble-2, on average, outperform all the individual models in the ensemble—particularly Ensemble-2, which is an ensemble of heterogeneous VLMs, achieves approximately a 1.4% improvement over FLAVA. While no single model achieved the best performance across all subsets of SugarCrepe++, the ensembled models either achieved the best performance or performed comparably to the best across all subsets. However, simply combining all available models, as in the Ensemble-All model, leads to a degradation in performance across all subsets of SugarCrepe++.

---

> > ### Author Rebuttal · Authors · 2024-08-20
> >
> > **Ensemble of Prompts:** In this approach, we combine the outputs of the same models when prompted with multiple augmentations of the same prompt. We applied this method to analyze the GPT-4o and BakLLaVA models. These results build upon the response to Reviewer 1 Question 3, which discusses prompt consistency in VLMs—here, we demonstrate that VLMs are highly sensitive to prompts and exhibit significant differences in performance, even with paraphrases of the same prompt. Table a1nX.2 shows the performance of BakLLaVA models when the outputs of three different prompts (which involve simple reordering of the captions) are combined. The ensemble of prompts approach does not achieve the best performance, likely due to the inconsistency of the model outputs in response to simple lexical changes.
> > Similar observations can be made from Table a1nX.3, which shows the performance of GPT-4o on SugarCrepe++ when using the ensemble of prompts approach.
> >
> > [1] LLM-BLENDER: Ensembling LLMs with Pairwise Ranking and Generative Fusion, ACL 2023.
> >
> > [2] SummaReranker: A multi-task mixture-of-experts re-ranking framework for abstractive summarization, ACL 2022.
> >
> > [3] Calibrating Language Models via Augmented Prompt Ensembles, ICML 2023.
> >
> > ***
> >
> > We hope we have addressed all your concerns and look forward to discussing any outstanding concerns in the remaining rebuttal discussion period.

---

> > > ### Comment · Reviewer_a1nX · 2024-08-23
> > >
> > > Thank you for all the clarifications and revision to the paper. I think my original concerns are mostly addressed properly and I would like to raise my rating from 6 to 7 and keep my confidence score unchanged.

---

> > > > ### Author Response · Authors · 2024-08-24
> > > > **Thank you**
> > > >
> > > > Thank you for considering our rebuttal and for engaging with us during the rebuttal period. We appreciate your invaluable insights and suggestions; they have improved the quality of our draft and expanded the applications of our dataset for evaluating consistency of generative VLMs.

---

### Official Review · Reviewer_hjnT · 2024-07-24
**Innovative SUGARCREPE++ dataset enhances evaluation of language models' semantic and lexical sensitivity**

**Rating:** 8
**Confidence:** 4

**Review:**

This paper is high-quality and clear, introducing the innovative SUGARCREPE++ dataset that fills a crucial gap in evaluating language models. The comprehensive evaluation and detailed methodology are impressive. It highlights key limitations in current models, guiding future research. While it could include a wider variety of models and reduce reliance on human validation, its contributions are significant.

**Strengths:**

1. The idea of building the dataset is great. Using three captions allows for dual-mode evaluation of Vision-Language Models (VLMs), and it effectively tests the models' multimodal and unimodal capabilities.
2. The experiments are quite thorough, as they evaluate models trained in different ways and with different structures, providing insights into the sources of the models' abilities.
3. The dataset generation and validation process is clearly described, including automated and human validation steps, ensuring high quality.

**Additional Feedback:**

None.

**Clarity:**

The paper is well-written and clearly structured. The methodology, dataset generation process, and experimental results are described in detail, clear to understand the research's scope and significance.

**Correctness:**

The methodology and experimental results presented in the paper are sound and well-supported by thorough evaluations. The conclusions drawn are consistent with the data and highlight important areas for improvement in current language models.

**Documentation:**

The documentation for this paper is clear and well-organized. It provides detailed explanations of the dataset generation process, validation steps, and experimental setup. The inclusion of both automated and human validation processes enhances the credibility and reliability of the dataset. The paper also offers comprehensive descriptions of the models evaluated and the specific metrics used, making it easy to follow the methodology and replicate the experiments. Overall, the documentation is thorough and supports the reproducibility of the research.

**Ethics:**

It uses publicly available data, respecting privacy norms. However, a bit more discussion on potential biases and their mitigation would be beneficial. Overall, the ethical standards are well-maintained.

**Limitations:**

Same as Opportunities For Improvement.

**Opportunities For Improvement:**

1. Lacks exploration of the dataset's own distribution, such as the distribution of topics in the text and images and statistics like the text's Vera score and grammar score.
2. The paper comparing the performance of different models on the current dataset. However, it lacks a comparison of these models' performance on other datasets, which would better demonstrate the effectiveness of the dataset.

**Relation To Prior Work:**

This paper takes benchmarks like MS-COCO and the original SUGARCREPE to the next level. SUGARCREPE++ fixes their issues by using triplet captions to better test how models handle semantic and lexical changes. It shows how this new dataset can improve language model evaluations and push the field forward.

**Summary And Contributions:**

The paper introduces the SUGARCREPE++ dataset to analyze the sensitivity of vision-and-language models (VLMs) and unimodal language models (ULMs) to semantic and lexical changes. Each dataset sample includes an image and three captions: two semantically equivalent but lexically different positives and one hard negative. This dataset aims to address gaps in evaluating semantic understanding in language models by considering lexical composition.

---

> ### Author Rebuttal · Authors · 2024-08-20
>
> We appreciate your insightful reviews and positive evaluation of our work. We are pleased to note that you liked our presentation, comprehensive analysis, and feel that our new dataset fills a crucial gap in evaluating language models. We agree with you that semantic equivalence understanding is indeed an important area for improvement in current language models. In the following response, we aim to address your concerns.
>
> ***(Q1) Lacks exploration of the dataset's own distribution, such as the distribution of topics in the text and images and statistics like the text's Vera score and grammar score.***
>
> We generated distribution plots of the topics in our dataset and conducted an analysis of the Vera and grammar scores. We will include these details in the final revision.
>
> We use images from MS-COCO [1,2], which defines 91 "stuff" categories (amorphous, uncountable elements like "sky," "water") and 80 "object" categories (countable, distinct items like "car," "dog"), which are fully covered in SugarCrepe++. We provide the distribution of these categories and their super-categories and subcategories in Figure hjnT.1 (in the PDF) . Additionally, we identify key topics in the caption text using the BERTopic [3] package, and Figure hjnT.2 (in the PDF) illustrates the top four topics across different subsets of SugarCrepe++.
>
> Hsieh et al [4] proposed an experiment based on Vera and Grammar scores to determine if language biases (e.g., fluency/grammar) can be exploited to “solve” the image-text matching task. We repeated this experiment on SugarCrepe++ and visualized the score gaps (e.g., 0.5*(Vera(P1)+Vera(P2)) - Vera(N)) in Figure hjnT.3 (in the PDF). For an ideal dataset that cannot be solved by exploiting language bias, we should observe that both Grammar/Vera score gaps are centered around zero. Accordingly, we confirm that both SugarCrepe++ as well as SugarCrepe cannot be solved by exploiting language biases. We also provide numerical scores in Table hjnT.4.
>
> ***
>
> ***(Q2) The paper comparing the performance of different models on the current dataset. However, it lacks a comparison of these models' performance on other datasets, which would better demonstrate the effectiveness of the dataset.***
>
> We thank the reviewer for their suggestion to compare SugarCrepe++ to other datasets. Recently, several benchmarks have been introduced to evaluate various capabilities of vision-language models (VLMs), including visual-question answering, conceptual understanding, visio-linguistic reasoning, and compositionality. Since evaluating compositionality is closely related to our work, we will provide a brief comparison between SugarCrepe++ and existing compositionality datasets such as ARO [5], VL-CheckList [6], Crepe [7], and SugarCrepe [4]. All of these previous datasets consist of an image, a single positive caption, and one or more negative captions. The distinction between these datasets is based on the type of negative captions, as explained below.
>
> *ARO:* The ARO dataset consists of “Swap” and “Shuffle” hard negatives. Swap hard negatives are created by replacing two words in the positive captions with different words, while Shuffle hard negatives are generated by rearranging two words in the positive captions.
>
> *VL-CheckList:* The VL-CheckList dataset consists of “Replace” hard negatives, which are generated by substituting the atomic components of the positive captions with other alternatives. Based on the type of atomic component replaced, the hard negatives are categorized into different types, such as object, attribute, and relationship.
>
> *SugarCrepe:* The SugarCrepe dataset consists of Swap, Replace, and Add hard negatives. These hard negatives are generated to be sensical and grammatically correct, in order to avoid the linguistic biases present in the ARO and VL-CheckList datasets.
>
> SugarCrepe++ builds upon the SugarCrepe dataset by introducing a new positive caption that is semantically similar but lexically different from the original positive caption. The triplet setting in SugarCrepe++ enables us to evaluate the ability of VLMs (in both unimodal and multimodal settings) to understand their capability to correctly encode semantics regardless of the lexical/syntactic properties of the sentence.
>
> Table hjnT.1 provides a comparison between previous datasets and SugarCrepe++ in terms of model performance. Vera and Grammar are blind models that do not have access to the images, while VLMs refer to the average performance of various CLIP variants, as detailed in Table 9 of the main paper.
>
> Datasets prior to SugarCrepe, such as ARO and VL-CheckList, were found to have linguistic biases due to nonsensical and non-fluent artifacts in their negative captions. These artifacts allowed blind models like Vera and Grammar, which do not have access to images, to outperform VLMs on the ARO and VL-CheckList datasets. SugarCrepe addressed this issue by introducing hard negatives that are free from these artifacts, enabling it to assess VLMs' true ability to understand the fine-grained atomic concepts in an image and identify the positive caption. Similar to SugarCrepe, VLMs perform better than Vera and Grammar on SugarCrepe++, indicating that the additional second positive caption in SugarCrepe++ does not introduce any linguistic biases (see Figure hjnT.3 and Table hjnT.4 in the attached PDF).
>
> ***
> [1] Microsoft coco: Common objects in context. ECCV, 2014.
>
> [2] Coco-stuff: Thing and stuff classes in context. CVPR, 2018.
>
> [3] BERTopic: Neural topic modeling with a class-based TF-IDF procedure. arXiv 2022.
>
> [4] Sugarcrepe: Fixing hackable benchmarks for vision-language compositionality. NeurIPS, 2023.
>
> [5] When and why vision-language models behave like bags-of-words, and what to do about it?. ICLR 2023.
>
> [6] Vl-checklist: Evaluating pre-trained vision-language models with objects, attributes and relations. arXiv 2022.
>
> [7] Crepe: Can vision-language foundation models reason compositionally?. CVPR 2023.

---

> > ### Author Rebuttal · Authors · 2024-08-20
> >
> > We hope we have addressed all your concerns and look forward to discussing any outstanding concerns in the remaining rebuttal discussion period.

---

### Official Review · Reviewer_DXNt · 2024-08-02
**Review of paper 1486**

**Rating:** 7
**Confidence:** 4
**Correctness:** Yes they are.
**Clarity:** Yes it is.

**Review:**

The paper writing is clear and easy to follow. The originality and significance of this work is decent.

Pros:
* The dataset includes 5 types of common replacements/swapping involving object, attributes and relations. This can help the community understand the performance breakdown under different scenarios.
* The dataset construction methodology with LLM generation and validation is reasonable and sound.
* The comprehensive evaluation is very helpful to understand how different existing VLMs behave on this benchmark with regards to their pretraining objectives, model/data size, and learning techniques.

Cons:
* It seems that this paper only evaluates the VLMs and ULMs in the zero-shot settings. How about reporting some baselines on few-shot and finetuning settings to expand the application of this dataset?
* I understand the strong legacy of MS-COCO dataset on the community. However, I believe we should move towards more diverse/large scale evaluation to assess the capability of VLMs better. It'd be nice to discuss the possibility of scaling up this type of semantic/lexical robustness evaluation moving forward.
* I'm wondering how sensitive is the performance of a generative VLM e.g. FLAVA/BLIP on SugarCrepe++ when you change the prompt. In other words, can proper prompting help mitigate the robustness issue to some extent?

**Strengths:**

See pros in the review.

**Additional Feedback:**

N/A.

**Documentation:**

Yes there are more details in the supp. materials.

**Ethics:**

No I don't think so.

**Limitations:**

Not that I'm aware of, but I think this dataset should be relatively safe for evaluation only.

**Opportunities For Improvement:**

See Cons in the review.

**Relation To Prior Work:**

yes it is.

**Summary And Contributions:**

The authors propose a dataset SugarCrepe++ which extends the idea of SugarCrepe to probe the semantic and lexical understanding capability of vision and language models. The dataset is more challenging than SugarCrepe and serves as a better testbed for the robustness of vision and language models.

---

> ### Author Rebuttal · Authors · 2024-08-20
>
> We appreciate your thorough review and insightful suggestions for expanding the applications of our dataset. Thank you for recognizing the significance of our dataset, appreciating our dataset construction methodology, and acknowledging our comprehensive analysis. We agree with you that it is indeed important to understand the performance breakdown of VLMs under different scenarios. In the following response, we aim to address your remaining concerns.
>
> ***
>
> ***(Q1) It seems that this paper only evaluates the VLMs and ULMs in the zero-shot settings. How about reporting some baselines on few-shot and fine-tuning settings to expand the application of this dataset?***
>
> Thank you for the insightful suggestion. We first give an overview of our experimental results, and then provide more detail:
>
> **Overview:** Full fine-tuning of the model, in some cases, leads to a degradation in performance compared to frozen model, possibly due to catastrophic forgetting [1]. To address this, we also report results using low-rank adapters (LoRA) [2], which consistently improved performance in all cases. Our initial few-shot learning results show significant performance degradation, potentially due to recency bias [3] or the choice and ordering of demonstration (reference) samples used in in-context learning. Detailed experimental results are provided below.
>
> **Fine-tuning results:**
> Table DXNt.1 (in attached PDF) presents the fine-tuning results. We fine-tuned the models using objective function employed in training sentence transformers [1] to ensure a fair comparison between the fine-tuned and zero-shot evaluations of the models. During inference, we adhered to the approach described in our paper. Experimental results show that while there are performance improvements on some subsets, fine-tuning can lead to performance degradation on others, which may be attributed to catastrophic forgetting [2]. To address this issue, we used LoRA [3]. Results demonstrate that LoRA improved performance across all subsets.
>
>
> **Few-shot results:**
> Table DXNt.2 provides the performance of GPT-4o under a few-shot setting (4/8 shot). For these experiments, we provided demonstration samples (4-shot and 8-shot) from the same subset  in addition to the query prompt (we use prompt-3 in Table 1 of Supplementary material).
> We observed significant degradation in performance of GPT-4o in both 4-shot and 8-shot settings. We suspect this could be due to the choice of the reference samples or due to recency effects  i.e., position of negative caption in the last reference sample influencing the decision of the VLM [4].
> We also observed that GPT-4o is very sensitive to the type of examples given as reference. When samples from other subsets are provided as reference samples, further degradation in performance was observed. Future research can explore the options of tuning the prompt and to find the set of reference samples which could make few-shot learning effective on our dataset.
>
> Although we conducted experiments with fine-tuning (noting improvements in performance using LoRA) and few-shot learning, we want to emphasize the importance of the zero-shot evaluation procedure on SugarCrepe++, as outlined in the paper. The main objective of our dataset is to assess a fundamental property of VLMs: whether these models can correctly encode semantics regardless of the lexical/syntactic properties of the sentence. We believe this capability should be inherently learned by the models during the pre-training process. Fine-tuning on our dataset may enhance performance on our task but could also lead to catastrophic forgetting, impacting the model's performance on other downstream tasks.
>
> We will include these additional results in the final revision.
>
>
> [1] Sentence-BERT: Sentence Embeddings using Siamese BERT-Networks, EMNLP 2019
>
> [2] An Empirical Study of Catastrophic Forgetting in LLMs During Continual Fine-tuning, arXiv, 2023
>
> [3] LoRA: Low-Rank Adaptation of Large Language Models, ICLR 2022
>
> [4] Calibrate before use: Improving few-shot performance of LLMs. ICML 2021
>
> ***
>
> ***(Q2) I understand the strong legacy of MS-COCO dataset on the community. However, I believe we should move towards more diverse/large scale evaluation to assess the capability of VLMs better. It'd be nice to discuss the possibility of scaling up this type of semantic/lexical robustness evaluation moving forward.***
>
> We agree with the reviewer that there is a necessity to build datasets outside the MS-COCO setting. Our approach is agnostic to the dataset and can be easily scaled to generate large datasets outside the MS-COCO format. Having said that, generating paraphrases of the captions that preserve the semantics while also accurately describing the image is a challenging task, even for the current state-of-the-art LLMs. As discussed in the paper, the outputs generated by the LLMs have several issues (examples provided in Section B.4 of Appendix) that include hallucination, incorrect spatial setting, etc. We found that the samples generated by the LLMs needed to be carefully evaluated by humans for quality checking. Even in the preparation of our dataset, this was a very time-consuming step and required a  lot of effort. This is a valuable discussion for future work to expand on our dataset, and will include it in the final revision.
>
> ***

---

> > ### Author Rebuttal · Authors · 2024-08-20
> >
> > ***(Q3) I’m wondering how sensitive is the performance of a generative VLM e.g. FLAVA/BLIP on SugarCrepe++ when you change the prompt. In other words, can proper prompting help mitigate the robustness issue to some extent?***
> >
> > Interesting question! This motivated us to explore additional issues with VLMs, and to further highlight the importance of our dataset in this context. We performed two different experiments to evaluate prompt sensitivity of generative VLMs using SugarCrepe++.
> >
> > **1) Paraphrasing of prompts** Here we prompt BLIP (in VQA format) with paraphrases of the same prompt (see Table DXNt.3 in PDF for prompt details), and report the performance in Table DXNt.3. It can be observed that for different paraphrases of the same prompt, there are huge differences in model performance, for both Prompt 1 and Prompt 2. Moreover, there is no single variant of the prompt that achieved best performance across all subsets of SugarCrepe++.
> >
> > Similar observations about prompt sensitivity are made for GPT-4o (see Section 1.1 and Table 12 in Supplementary material), where we show significant differences in performance of GPT-4o by prompting with different versions of the same prompt.
> >
> >
> > **2) Reordering options in the prompts:** As BLIP cannot handle long prompts, we analyzed BakLLaVA [1], a generative VLM based on Mistral 7B augmented with LLaVA architecture. We prompt BakLLaVA using the prompts shown in Figure DXNt.1,  and report results on SugarCrepe++  in Table DXNt.4. We observed that by simply changing the ordering of the captions, the performance of the BakLLaVA model changes drastically. **This sensitivity of model performance to simple paraphrases of the prompt further highlights the importance of using our SugarCrepe++ dataset to evaluate VLMs ability to understand semantic and lexical alterations.**
> >
> > We believe that generative VLMs that do not perform well on the SugarCrepe++ dataset might potentially struggle with understanding prompts that have different lexical arrangements but convey the same meaning, thereby underscoring the value and applicability of our dataset.
> >
> > [1]  Improved Baselines with Visual Instruction Tuning, CVPR, 2024
> > ***
> >
> > We hope we have addressed all your concerns and look forward to discussing any outstanding concerns in the remaining rebuttal discussion period.

---

> > > ### Author Response · Authors · 2024-09-01
> > > **Follow Up - Deadline Approaching**
> > >
> > > Thank you again for your insightful suggestions and detailed review recognizing the originality and significance of our work. Based on the reviewer's feedback, we explored applications of our dataset in fine-tuning, few-shot learning and prompt-sensitivity evaluation of generative VLMs. We feel that these new experiments are valuable additions to the paper and hope that we have been able to clarify all outstanding concerns through our above rebuttal response. Please let us know if there are any further questions/concerns and we will gladly address them before the discussion window closes.

---

> > > > ### Comment · Reviewer_DXNt · 2024-09-01
> > > >
> > > > Thank you authors very much for the comprehensive rebuttal with the experiments on finetuning, few-shot learning and prompt-sensitivity evaluation. All of these are very thought-provoking observations, which I believe would shed light on future research directions for the community. After carefully considering the rebuttal and other reviews, I decide to raise my score to 7: Accept. Thanks again!

---

### Author Rebuttal · Authors · 2024-08-20

We thank all the reviewers for their insightful and thoughtful reviews with suggestions for improvements. We are happy to see that the reviewers appreciated the novelty (DXNt, hjnT, a1nX, XGk7) of the proposed evaluation technique and the importance (DXNt, hjnT, a1nX) of our new dataset and found the paper to be well-presented (DXNt, hjnT, a1nX) with extensive experiments and analysis (DXNt, hjnT, a1nX).

**Additional Experiments** Based on the reviews, we were excited to see that our new dataset and evaluation technique prompted interesting ideas for additional experiments. We conducted additional experiments during the rebuttal period and have provided the results in separate PDFs for each reviewer. To differentiate these new tables and figures from those in the paper, we adopt the following naming convention: Table/Figure {REVIEWER-CODE}.{NUMBER}. For example, Figure 1 in the PDF for reviewer DXNt is labeled Figure DXNt.1. We summarize the new experiments as follows.

[E1] **Fine-tuning on SugarCrepe++:** We evaluated performance improvements upon fine-tuning (including LoRA). [DXNt response/PDF]

[E2] **Few-shot Learning:** We evaluated GPT-4o on SugarCrepe++ by providing a few (4/8) demonstrations in the prompt. [DXNt response/PDF]

[E3] **Prompt Sensitivity Evaluation of generative VLM:** We demonstrate significant performance differences in generative VLMs (e.g., BLIP, BakLLaVA and GPT-4o) for different semantically-identical prompts. [DXNt response/PDF]

[E4] **Model Consistency Evaluation:** We evaluated ensembling as a type of consistency evaluation for generative VLMs and explored both model-ensembling as well as prompt-ensembling.  [a1nX response/PDF]

[E5] **Additional Dataset Statistics:** We discuss the image categories and topics covered in SugarCrepe++. We also analyze/visualize the distributions of Vera and Grammar scores. [hjnT response/PDF]

[E6] **VisualGPTScore/VQAScore with Generative VLMs:** We provide additional results on large generative VLMs using recent SOTA approaches.  [XGk7 response/PDF]

***
Reviewer XGk7 suggested that we provide definitions for the Lexical, Syntactic and Semantic properties along with a method for quantifying lexical/syntactic similarities. In the following, we provide definitions, describe a metric for measuring lexical/syntactic similarities and visualize the statistics of this metric applied to SugarCrepe++ dataset.

**Lexical, Syntactical and Semantic Properties:** First, we provide the Mariam-Webster definitions of “lexical”, “syntax” and “semantic”:
* **Lexical:** of or relating to words or the vocabulary of a language as distinguished from its grammar and construction

* **Syntax:** the way in which linguistic elements (such as words) are put together to form constituents (such as phrases or clauses)

* **Semantic:** of or relating to meaning in language

Together, lexical and syntactic aspects of the sentence refer to the choice of words and writing style used to construct a sentence. The meaning expressed by the sentence is referred to as semantics. By varying the lexical and syntactic aspects, we can create different sentences that express the same meaning (i.e., semantically equivalent). Likewise, we can create sentences that are semantically non-equivalent but are very close in terms of their lexical and syntactic aspects. By extending SugarCrepe, SugarCrepe++ consists of 3 captions for each image (P1, P2 & N) where P1 and P2 are correct captions (see Fig. 1 of our paper) and are related as follows:
* (P1,N): similar lexical/syntactic properties and semantically non-equivalent.

* (P1,P2): different lexical/syntactic properties and semantically equivalent.

* (P2,N): different lexical/syntactic properties and semantically non-equivalent.

**Measuring Lexical/Syntactic Variations:** Syntactic aspects of a sentence representing how words are put together to form constituents are best represented by its constituency parse trees. Accordingly, recent techniques, such as FastKASSIM [1] and CASSIM [3], utilize the constituency parse trees to estimate the syntactic similarity between a pair of sentences. On the other hand, lexical similarity between sentences can be measured by comparing the words used in each sentence and simple measures to compute lexical similarity include jaccard scores (i.e., word overlap) and edit-distances.

To address the reviewer’s request, we define the following metric to jointly measure the syntactic and lexical similarity (SLS) between two sentences $U$ and $V$:

SLS$(U,V)$ $=$  LexicalSimilarity$(U, V)$ $\times$ SyntacticSimilarity$(U, V)$

We use the normalized levenshtein similarity based on [2] as a measure of lexical similarity and FastKASSIM [1] with default parameters as a measure of syntactic similarity. We note that both of these metrics yield similarity scores in the range of $[0,1]$ and hence SLS$\in [0,1]$ closer to 1 indicates higher syntactic and lexical similarity.

For completeness, we extended this analysis and include a boxplot of the SLS in Figure XGk7.1 of the attached PDF. We find that the $(P_1, P_2)$ sentence pairs have consistently lower SLS scores as compared to $(P_1, N)$ sentence pairs: specifically, on average, we find that majority of the (P1, P2) similarity scores lie between 0.2-0.4 while majority of the (P1, N) scores are greater than 0.75. In summary, this shows that (a) the sentences which we intended to be lexically/syntactically similar are indeed so, according to this measure; and (b) the sentences which we intended to be lexically/syntactically different are indeed so as well, according to this measure.

[1] FastKASSIM: A Fast Tree Kernel-Based Syntactic Similarity  Metric.  EACL 2023.

[2] A normalized Levenshtein distance metric. TPAMI 2007.

[3] Conversation Level Syntax Similarity Metric. Behavior Research Methods, 2017.

***

---

> ### Comment · Reviewer_XGk7 · 2024-08-20
> **Is there an individual rebuttal?**
>
> Thanks for providing this overall rebuttal. I am however not seeing the responses to individual reviewers.
>
> Edited: Sorry -- I realized that you just posted the rebuttal 10 minutes ago. I will wait for the individual response.

---

> > ### Author Rebuttal · Authors · 2024-08-20
> >
> > Dear XGk7, thank you very much for your prompt response! We will be posting individual rebuttals over the next several minutes. We look forward to discussing further with you!

---

### Decision · Program_Chairs · 2024-09-26

**Decision:**

Accept (Poster)

**Comment:**

I would like to acknowledge the reviewers and authors for their deep engagement during the discussion period. Overall, there is a clear signal that this work constitutes a valuable contribution to the NeurIPS Datasets and Benchmarks track. It proposes a well-designed benchmark targeted towards assessing the lexical/semantic sensitivity of both multimodal and unimodal models, and conducts an assessment of a large number of recent models in these classes.

As highlighted by reviewers [a1nX, hjnT, DXNt], the paper is well-written, with a good level of detail and easy to follow. I appreciate that initial concerns from reviewers (in particular [XGk7]) around precision of the "lexical" and "semantic" terms were addressed in detail by the authors during the discussion period, so I trust the revised paper will include these clarifications. The proposed dataset is of relevance to the community and the construction methodology seems sound [DXNt, hjnT], especially given the involvement of human review. The subsets within the dataset with different lexical variations ("Swap Object", "Swap Attribute", etc.) are of particular value [DXNt], however I am unable to find the precise definition of these subsets (did they use different instruction prompts, or examples?) so would request these details be added or more clearly highlighted in the revision. The evaluation of different approaches using SUGARCREPE++ is comprehensive [DXNt, hjnT, a1nX], with a valuable comparison to SUGARCREPE [a1nX], and the authors added additional scenarios during the discussion period (considering few-shot/fine-tuning [DXNt], adding generative VLMs [DXNt, XGk7], adding more details on the distribution of the dataset itself and comparison to related datasets [hjnT]), which I expect to see reflected in the final version where possible.

Overall my assessment is that this is solid work which fits well in the NeurIPS Datasets and Benchmarks track.